# A megaplasmid family driving dissemination of multidrug resistance in *Pseudomonas*

Adrian Cazares [1✉], Matthew P. Moore[1], James P. J. Hall [2], Laura L. Wright[1], Macauley Grimes[1], Jean-Guillaume Emond-Rhéault[3], Pisut Pongchaikul [4], Pitak Santanirand[4], Roger C. Levesque [3], Joanne L. Fothergill[1] & Craig Winstanley[1✉]

Multidrug resistance (MDR) represents a global threat to health. Here, we used whole genome sequencing to characterise *Pseudomonas aeruginosa* MDR clinical isolates from a hospital in Thailand. Using long-read sequence data we obtained complete sequences of two closely related megaplasmids (>420 kb) carrying large arrays of antibiotic resistance genes located in discrete, complex and dynamic resistance regions, and revealing evidence of extensive duplication and recombination events. A comprehensive pangenomic and phylogenomic analysis indicates that: 1) these large plasmids comprise an emerging family present in different members of the *Pseudomonas* genus, and associated with multiple sources (geographical, clinical or environmental); 2) the megaplasmids encode diverse niche-adaptive accessory traits, including multidrug resistance; 3) the accessory genome of the megaplasmid family is highly flexible and diverse. The history of the megaplasmid family, inferred from our analysis of the available database, suggests that members carrying multiple resistance genes date back to at least the 1970s.

[1] Institute of Infection and Global Health, University of Liverpool, Liverpool, UK. [2] Department of Evolution, Ecology and Behaviour, University of Liverpool, Liverpool, UK. [3] Institute for Integrative and Systems Biology (IBIS), University Laval, Quebec City, QC, Canada. [4] Ramathibodi Hospital, Mahidol University, Bangkok, Thailand. ✉email: A.Cazares-Lopez@liverpool.ac.uk; C.Winstanley@liv.ac.uk

                    1

The spread of antimicrobial resistance (AMR) is recognised as a key global challenge to human health[1]. The so-called ESKAPE pathogens (*Enterococcus faecium, Staphylococcus aureus, Klebsiella pneumoniae, Acinetobacter baumannii, Pseudomonas aeruginosa* and *Enterobacter* sp.) are the leading cause of nosocomial infections worldwide[2]. In particular, *P. aeruginosa* causes a wide range of opportunistic infections[3], is often associated with multidrug resistance (MDR)[4] and has been highlighted by the World Health Organisation as a critical (Priority 1) pathogen associated with AMR[5].

The rapid spread of resistance to different host backgrounds is often facilitated by mobile genetic elements (MGE): entities that are adapted to moving DNA between replicons and cells[6], including insertion sequences (ISs), transposons, integrons and conjugative plasmids[7]. In contrast to pathogens from the enterobacteriaceae, the role of plasmids in the spread of MDR in *P. aeruginosa* is not well understood.

It is recognised that AMR can develop and spread rapidly in regions where antibiotics are inappropriately sold/used and freely available and that hospital settings provide excellent opportunities for the dissemination of resistance. In Thailand, carbapenem-resistant *P. aeruginosa* is an increasing problem, especially in the context of hospital-acquired infections[8]. Previous studies have sought to characterise the genetic basis underlying carbapenem resistance among *P. aeruginosa* clinical isolates in Thailand[9], and globally[10], but the role of plasmids is poorly understood.

Whole genome sequencing can be a powerful epidemiological tool in diagnostic and public health microbiology[11], but it has limitations that have restricted our understanding of the role of MGE. The use of short-read data from the most popular high-throughput platforms makes the assembly of plasmids, which often contain repetitive sequences, difficult[12]. Long read sequencing, using platforms such as PacBio, can resolve the status of plasmids, allowing comparative analysis that can elucidate the routes by which AMR genes spread between lineages[13].

Here we describe a combination of long-read and short-read genome sequencing that allowed us to characterise a family of megaplasmids contributing to the spread of MDR among *P. aeruginosa* in a hospital in Thailand. By surveying sequence databases, previously overlooked members of this megaplasmid family were identified in isolates from multiple sources from around the world, including the environment, and in non-*aeruginosa* species of *Pseudomonas*. Related plasmids have a shared core genome but vary in the carriage of AMR genes, identifying the megaplasmid as a vehicle for the consolidation and dissemination of resistance. Our study provides valuable insights into how MDR plasmids emerge from environmental reservoirs into a clinical setting.

## Results

**Antimicrobial susceptibilities of clinical isolates**. We assembled a collection of 48 *P. aeruginosa* isolates associated with different kinds of infection and carried out susceptibility tests for five antibiotics representing different modes of action (Supplementary Data 1). Nine of the 48 isolates were resistant (either full or intermediate) to all 5 antibiotics, while a further 6 were resistant to 4 of the 5 antibiotics.

**Identification of related MDR megaplasmids**. We selected three of the most resistant isolates (2101, 2436 and 4068) for long-read (PacBio) genome sequencing. Genomes of the strains 2436 and 2101 were assembled into two complete circularised contigs. In all, 2436 and 2101 chromosomes are 6,782,092 and 6,573,638-bp long and feature 6214 and 6041 protein-coding genes,

respectively. Both isolates also carried related megaplasmids (named pBT2436 [423 kb] and pBT2101 [440 kb], respectively), harbouring multiple AMR genes. A pairwise comparison of pBT2436 and pBT2101 indicated that these plasmids shared approximately 90% of their genome, differing mostly in the regions carrying AMR genes (resistance regions RR1 and RR2; Supplementary Fig. 1). In addition, there was a 7.7-kb region (Variable Region VR1) present in pBT2101 but absent from pBT2436, which was not related to AMR but was also present in the chromosome of the isolate 4068. The annotations of pBT2436 and pBT2101 identified shared regions relating to replication and partitioning (*repA*, *parAB*), DNA transfer (*traGBV*, *dnaG* and type IV pilus-related/type II secretion genes), heavy metal resistance (*terZABCDEF*) and chemotaxis (*che* genes) (Supplementary Data 2). Although the two plasmids harboured *mer* operons, these differed both in gene content and sequence identity at the nucleotide level (*merRTPCA* in pBT2436 and *merRTPFADE* in pBT2101, which contains additional but divergent *merRTP*; Fig. 1). The genome of the strain 4068 was assembled into five contigs, one of which corresponded to a complete circularised plasmid of 51 kb with no identifiable AMR genes, therefore it was not analysed further in this study.

**AMR regions in pBT2436 and pBT2101 are mosaic and dynamic**. Both plasmids carried an extensive array of AMR genes, encoding resistances against a wide range of antibiotics including beta-lactams (*VEB, blaOXA-10, CARB-3*), aminoglycosides (*ANT, APH* and *aad* genes), sulphonamides (*sul1*), tetracyclines (*tet* genes), macrolides (*ermE*) and phenicols (*floR*). We also identified genes for an efflux pump (*mexCD-oprJ*) present in pBT2436 (Fig. 1).

In pBT2436, there was one large (RR1, 57.1 kb) and one small (RR2, 15.7 kb) resistance region containing AMR genes. In pBT2101 there was a single, large resistance region (RR1, 81.6 kb) (Fig. 1). By aligning each of the two larger resistance regions with themselves, we identified both large and small duplicated regions, including a 35.9-kb region duplicated but inverted in pBT2101, with a unique central region of 4.36 kb in between the two duplicated areas containing a *mer* operon (Fig. 1; Supplementary Fig. 2).

The resistance regions were all rich in transposases and integrases, both unique and shared, in proximity to AMR genes (Fig. 1). Some regions were shared between the two large resistance regions of pBT2436 and pBT2101 (e.g. *sul1-floR-tetR-tetG*). In other areas, there were partial matches where rearrangements have shuffled the order of genes. For example, in pBT2101 there was a cluster of resistance genes *blaOXA-10-aadA-VEB-1-ANT(2″)-Ia*, whereas in pBT2436, the genes were found in the order *VEB-2-ANT(2″)-Ia-arr2-cmlA5-blaOXA-10-aadA*.

Within individual resistance regions, there were also examples of both direct and inverted repeats (e.g. *tnpA_3* and *tnpA_4* and *sul1_1* and *sul1_2* in pBT2436), ranging in size from 48 to 1635 bp. Most of the repeats were shared between the resistance regions of the two plasmids but they differed in their arrangements (Fig. 1).

There were examples of transposon insertions contributing to divergence between regions shared by the two megaplasmids. For example, the genes APH(6)-Id_1/2 in pBT2101 and *tnpA_1* in pBT2436 represent truncated versions of their homologues due to the presence of adjacent *tnpA_1/10* and *tnpA_2* transposase genes, respectively (Fig. 1).

We searched for ISs in the megaplasmid resistance regions to gain insights about the origin of the transposase, integrase and resistance genes. Matches to eight different IS were identified

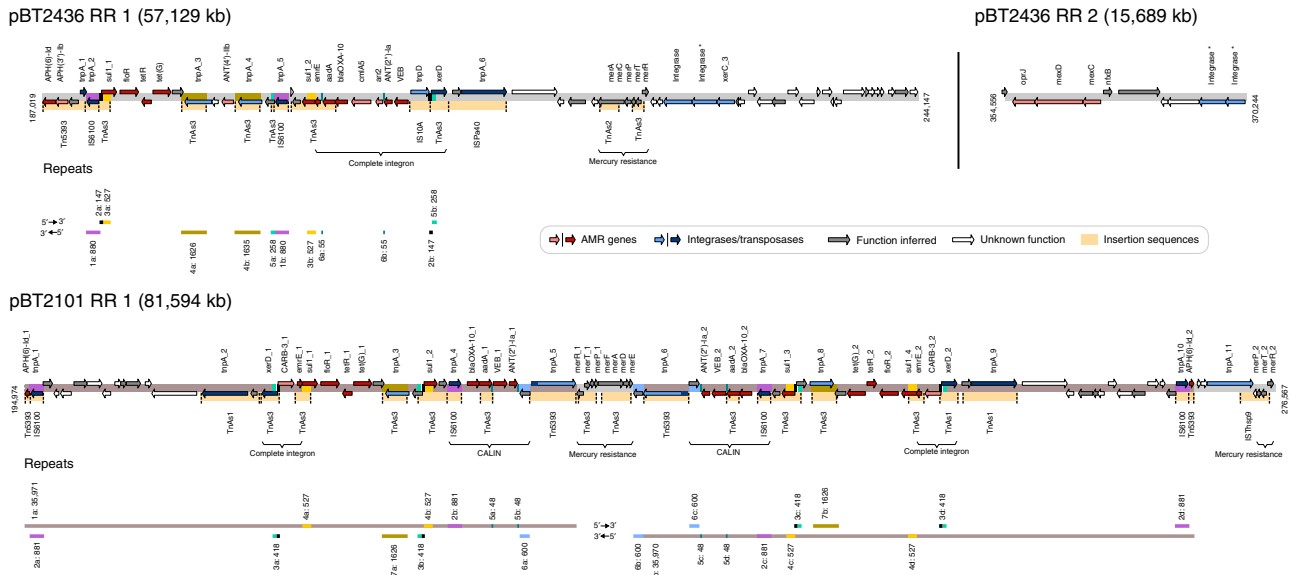

**Fig. 1 Maps of the pBT2436 and pBT2101 AMR regions.** The coordinates of the AMR regions within the megaplasmid sequences are shown flanking the maps. ORFs and repeats, represented as coloured arrows and blocks, respectively, are indicated in the maps along with the location of the detected mercury resistance operons and complete integrons and clusters of *attC* sites lacking integron-integrases (CALIN) as identified by the Integron Finder tool[61]. Insertion sequences (ISs) recognised with ISfinder[62] are represented as orange blocks in the maps along with their corresponding names. Different contiguous IS identified within a plasmid region are separated by dotted lines. Maps are drawn to scale. Gene names or products of selected ORFs of interest are indicated above their corresponding arrows. Gene duplications occurring within the same AMR region are numbered and distinguished with an "_N" suffix in their names. Darker shade-coloured arrows denote ORFs shared by the two megaplasmids' RR1 regions. The different identified repeats are numbered according to their first occurrence in the AMR region and a suffix letter (a–e) is included to distinguish between them. Repeat size in base pairs is displayed next to their corresponding blocks. Repeat blocks sharing matching colours between the two megaplasmids' RR1 regions represent shared regions.

scattered across the resistance regions, including some within resistance genes, indicating a key role for IS in the acquisition of resistance from different origins. Matches corresponding to Tn5393, IS6100 and TnAs3 were shared by the two megaplasmids (Fig. 1, Supplementary Data 2). Although the origin of the recognised IS is diverse, most of the matches in pBT2436 and pBT2101 corresponded to elements described in *Aeromonas salmonicida*. No matches to known IS were detected in the pBT2436 RR2.

In addition to predicted integrases, RR2 in plasmid pBT2436 carried genes for an efflux pump *mexC-mexD-oprJ* and a divergently transcribed *nfxB* gene, encoding the putative efflux pump repressor. Since this gene organisation is identical to that present in the chromosome of the reference strain PAO1, we compared RR2 to the chromosome of the pBT2436-carrier strain (Supplementary Fig. 3). Low levels of nucleotide sequence identity (at best 78% sequence identity to a region comprising only *mexD* and parts of *mexC/oprJ*) and differences in flanking genes suggested that the carrier strain chromosome was not the source of RR2. Searches of the wider database revealed closer matches to sequences of diverse taxonomic origins, including both chromosomes and plasmids. The best match corresponded to a region in the chromosome of *Aeromonas hydrophila* strain WCHAH045096 (Accession CP028568), with identity levels of 98% across the entire resistance region (Supplementary Fig. 3). Other close matches were found to regions of the non-related plasmids pBKPC18-1 from *Citrobacter freundii* (Accession CP022275) and pMKPA34-1 from *P. aeruginosa* (Accession MH547560) which mainly differed from the pBT2436 RR2 by the presence of genes associated with various IS, suggesting a role for these in its transmission (Supplementary Fig. 3). The dynamic nature of these resistance regions, in contrast to the general conservation across the rest of the plasmid backbone, suggests that resistance genes have been assembled independently in the different plasmid backgrounds.

**Distribution of related megaplasmids among clinical isolates.** In order to determine whether related plasmids were present in other *P. aeruginosa* from patients in the same hospital, 23 clinical isolates (including 2101 and 2436), chosen to represent a mixture of more resistant and less resistant isolates (Supplementary Data 1), were genome sequenced using the short-read Illumina platform. The multi-locus sequence-type (MLST) profiles extracted from the genomes and a kmer-based sequence comparison indicated that the isolates were highly diverse genetically (Supplementary Data 1 and Supplementary Fig. 4).

When Illumina reads were mapped on to the two plasmids as reference genomes, four of the genomes (including those of strains 2101 and 2436) mapped with coverage >90% (Supplementary Fig. 5), indicating the presence of related plasmids in isolates 3583 and 638, both of which also displayed extensive AMR (Supplementary Data 1). Isolates 2436 and 638, obtained from the same patient but from different types of sample (sputum and blood, respectively), shared the same sequence type (ST) (Supplementary Data 1) and their plasmids shared high levels of identity. Isolates 2101 and 3583 share the same sequence for six of the seven MLST loci but were isolated from different patients. Although the plasmids they carry share high levels of identity, when sequencing reads from isolate 3583 were mapped on to the pBT2101 genome, there was a clear reduction in numbers of reads mapping to the 81.6 kb RR1 containing the large duplication, consistent with the notion that the plasmid in isolate 3583 carries only one copy of the resistance genes rather than two (Supplementary Fig. 5). Likewise, when 2101 and 3583 sequencing reads were aligned to the pBT2436 sequence, only 2101 displayed

**Table 1 Complete megaplasmids of the pBT2436-like family.**

| Plasmid | Bacterial species | Strain | Size | Number of proteins | Country | Source | Accession number | Reference |
|---|---|---|---|---|---|---|---|---|
| pBT2436 | *P. aeruginosa* | 2436 | 422,811 | 536 | Thailand | Respiratory infection | CP039989 | This work |
| pBT2101 | *P. aeruginosa* | 2101 | 439,744 | 556 | Thailand | Respiratory infection | CP039991 | This work |
| unnamed2 | *P. aeruginosa* | AR_0356 | 438,531 | 556 | NA | NA | CP027170 | NA |
| unnamed2 | *P. aeruginosa* | AR439 | 437,392 | 545 | NA | NA | CP029096 | NA |
| unnamed3 | *P. aeruginosa* | AR441 | 438,529 | 556 | NA | NA | NZ_CP029094 | NA |
| pJB37 | *P. aeruginosa* | FFUP_PS_37 | 464,804 | 597 | Portugal | Respiratory infection | KY494864 | ref. [21] |
| pBM413 | *P. aeruginosa* | PA121617 | 423,017 | 542 | China | Respiratory infection | CP016215 | ref. [31] |
| pOZ176 | *P. aeruginosa* | PA96 | 500,839 | 623 | China | Respiratory infection | KC543497 | ref. [19] |
| p12939-OXA | *P. aeruginosa* | NA | 496,436 | 608 | China[a] | NA | MF344569 | NA |
| p727-IMP | *P. aeruginosa* | NA | 430,173 | 534 | China[a] | NA | MF344568 | NA |
| pA681-IMP | *P. aeruginosa* | NA | 397,519 | 486 | China[a] | NA | MF344570 | NA |
| pR31014-IMP | *P. aeruginosa* | NA | 374,000 | 456 | China[a] | NA | MF344571 | NA |
| pRBL16 | *P. citronellolis* | SJTE-3 | 370,338 | 485 | China | Wastewater sludge | CP015879 | ref. [35] |
| p1 | *P. koreensis* | P19E3 | 467,568 | 599 | Switzerland | *Origanum majorana* | CP027478 | ref. [17] |
| pSY153-MDR | *P. putida* | SY153 | 468,170 | 580 | China | Urinary tract infection | KY883660 | ref. [14] |

*NA* not available.
[a]Country of isolation is putative, based on the information of the submitter's country in GenBank.

twice the number of reads mapping to the pBT2436 resistance region (Supplementary Fig. 5).

A phylogeny of the Thai isolates shows that the megaplasmids can be found in strains from the two major *P. aeruginosa* groups (Supplementary Fig. 4). Isolates 2101 and 3583 are more closely related to each other than to any other isolate in the collection but they still display divergence compared to isolates 2436 and 638, obtained from the same patient. This is consistent with a transmission event linking the 2101 and 3583 isolates.

Extended antimicrobial susceptibility tests carried out on these four isolates identified some variations (Supplementary Table 1). For netilmicin, whereas for three of the isolates zones of inhibition were close to the European Committee on Antimicrobial Susceptibility Testing (EUCAST) breakpoint diameter of 12 mm (isolate 2101 just below and isolates 638 and 3583 just above), no zone of inhibition was detected for isolate 2436. When comparing the two strains carrying megaplasmids for which we obtained complete genomes, the only difference was that isolate 2101 was susceptible to amikacin.

**The family of pBT2436-like megaplasmids is widely distributed.** Having identified the presence of members of the same family of megaplasmids (which we refer to as the pBT2436-like megaplasmids) associated with MDR in four of the clinical isolates from Thailand, we carried out homology searches targeting complete sequences at the non-redundant NCBI nucleotide database. Details of the 13 additional megaplasmids identified, ranging in size from 370 to 501 kb and encoding from 456 to 623 proteins, are shown in Table 1.

Although ten of the megaplasmids were present in strains of *P. aeruginosa*, we identified related plasmids in three non-*aeruginosa* species of *Pseudomonas*: *P. putida*, *P. citronellolis* and *P. koreensis* (Table 1). The majority of isolates for which information about the country of isolation was available were linked with China and two of the three non-*aeruginosa* isolates were from sources other than human samples. The exception was the strain of *P. putida*, which was associated with a human urinary tract infection[14].

A comparative analysis of the 15 complete megaplasmid sequences revealed that they share high synteny and extensive sequence similarity, with the detected variation distributed at discrete *loci* across the genomes but mainly concentrated in large regions rich in AMR genes, transposases and integrase genes

(Fig. 2). The 7.7-kb variable region (VR1, Fig. 2) of pBT2101 was absent from any of the other members of the megaplasmids group. Further homology searches focusing on VR1 only identified one match in GenBank, corresponding to the chromosome of strain MRSN12280, recently described as the first report of a *P. aeruginosa* colistin-nonsusceptible isolate carrying the resistance gene *mcr*[15]. However, annotation of the 7.7-kb pBT2101 unique region suggests that this region does not contribute to resistance. The GC content was similar among the members of the megaplasmids family (range 55.9–57.6%) although clear deviations from the average were detected, mostly in the large variable regions (Fig. 2). In pBT2436 and pBT2101, GC- and AT-rich regions coincided with the location of multiple transposase, integrase and resistance genes predicted to be part of various IS with diverse taxonomic origins (Supplementary Fig. 1, Fig. 1). The GC content of the megaplasmids was lower than the median GC content reported for the genomes of their hosts, which range from 59.9% (*P. koreensis*) to 67% (*P. citronellolis*).

**Core and accessory genome of the pBT2436-like megaplasmids.** Based on the comparative analysis of the 15 members of the megaplasmid family, we identified a core genome consisting of 261 orthologous protein groups, including proteins with roles in plasmid replication and partitioning, plasmid transfer, heavy metal resistance, chemotaxis and a set of radical S-adenosyl-L-methionine (SAM) enzymes (Fig. 2; Supplementary Data 2). On average, approximately 48% of each megaplasmid comprised the core genome (range 42–57%). Pangenome analysis using two different approaches yielded a consensus of 1164 orthologous protein groups (Supplementary Fig. 6). Rarefaction and accumulation curves were indicative of an open pangenome with a well-defined core and suggested that the addition of more genomes was unlikely to impact much further on the size of the core genome (Fig. 3). Further analysis was carried out to distinguish between the strict core genome (present in all 15 megaplasmids) and other pangenome compartments[16] (Supplementary Fig. 6). The greatest number of gene clusters corresponded to unique, or nearly unique genes, but with numbers varying between the different megaplasmids, from 1 to 93 unique genes (Supplementary Fig. 6; Supplementary Data 3), the largest being found in a plasmid (p1) present in a *P. koreensis* isolate from healthy marjoram leaf material[17]. Genes unique to plasmid p1 include a large cluster related to copper resistance and catabolic genes for the

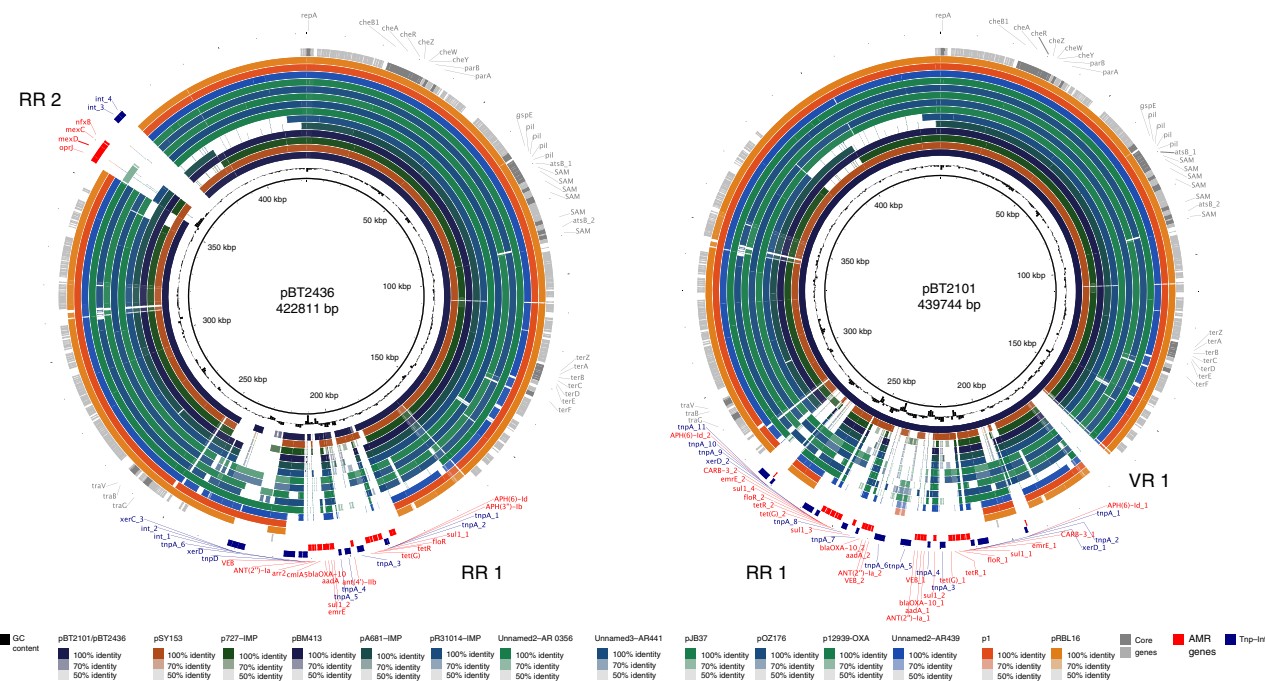

**Fig. 2 Genome comparison of megaplasmids of the pBT2436-like family.** Fourteen complete *Pseudomonas* megaplasmid sequences, indicated at the bottom of the figure and represented as colour rings, were aligned to the pBT2436 (left) and pBT2101 (right) genomes at the nucleotide level. Solid circles denote sequence homology to pBT2436/pBT2101, whereas gaps within the rings correspond to regions lacking sequence similarity. Innermost rings (black) represent the GC content deviation from the average in the reference genomes. The three outermost rings (from innermost to outermost) indicate the location in pBT2436/pBT2101 of the core genes identified from the pangenome analysis of the family (grey), AMR genes (red), and genes encoding integrases or transposases (blue). Names of genes of interest and the position of the pBT2436/pBT2101 AMR (RR) and variable (VR) regions are indicated in the figure.

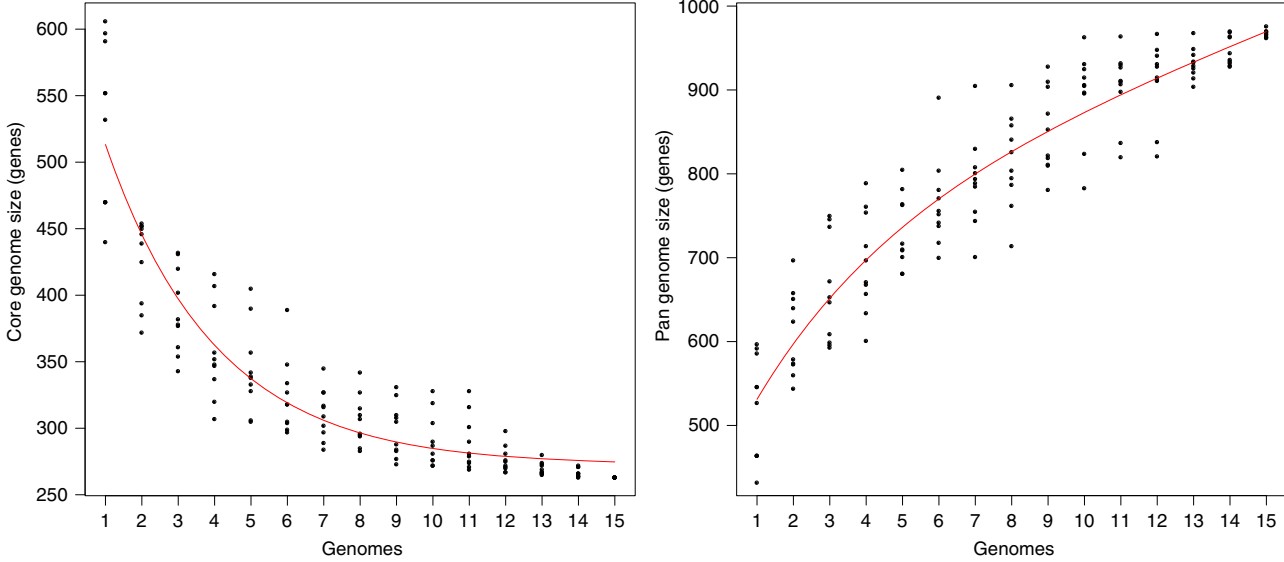

**Fig. 3 Rarefaction curves for the core and pangenome of the pBT2436-like megaplasmid family.** Graphs show the estimated size of the core (left) and pangenome (right) of the pBT2436-like group. Plotted data come from sampling experiments with ten random-seeded replicas from the BDBH-[16] (core) or OMLC-based[67] (pan) protein clustering. Fitted curves follow the Tettelin function[68].

conversion of 4-hydroxy benzoate (*pobA*, *phbH*) or vanillate (*vanA*, *vanB*, *iacF*, *vanK*) to protocatechuate, which can then be metabolised to central metabolism intermediates via the β-ketoadipate pathway. Vanillate, a lignin derivative, and 4-hydroxybenzoate (also known as para-hydroxybenzoate) are both plant-derived compounds.

In the plasmids hosted by *P. aeruginosa*, there were also clusters of "unique" genes, including a cluster of putative polysaccharide biosynthesis genes in the plasmid carried by *P. aeruginosa* AR439 and genes putatively involved in the oxidation of phosphite to phosphate in the plasmid p727-IMP. Other "unique" genes have putative functions related to antibiotic

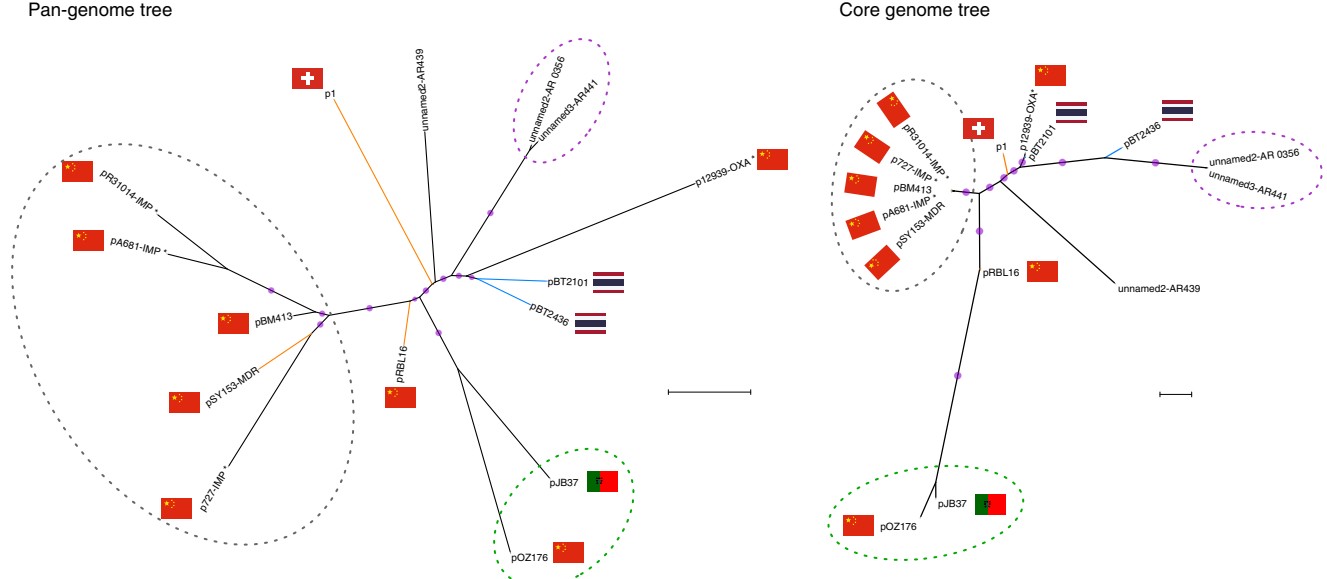

**Fig. 4 Phylogenomic analysis of complete megaplasmids of the pBT2436-like family.** Maximum-likelihood unrooted phylogenetic trees displaying the relationships among the megaplasmids reported in this work (blue branches) and 13 *Pseudomonas* homologous plasmids detected in GenBank. Branches corresponding to plasmids from non-*aeruginosa* species are coloured in orange. Pangenome phylogeny (left) was estimated from the pangenome matrix and reflects the evolutionary relationship of the megaplasmids in terms of their gene content, i.e. presence/absence patterns of the 1164 clusters contained in the matrix. The core genome phylogeny (right) was estimated from the concatenated set of 105 top-ranking alignments from core genes selected as phylogenetic markers and depicts the patterns of divergence of the megaplasmid genomic backbone. Taxa clustered together in both trees are highlighted with dotted-line ovals of matching colours. Purple dots represent approximate Bayesian posterior probability values >0.99 or >0.81 for the corresponding nodes in the core genome and pangenome trees, respectively. Scale bars correspond to 0.1 expected substitutions per site under the binary GTR+FO (pangenome tree) or best-fitting GTR+F+ASC+R2 (core genome tree) models. Where known, sources are indicated by nation flags. Asterisks in plasmid names denote cases where the geographical source is putative, based on the information of the submitter's country in GenBank.

resistance, conjugation, transposases/integrases and chemotaxis (Supplementary Fig. 7).

*Pseudomonas* plasmids can be classified according to incompatibility group[18]. One member of the megaplasmid family, plasmid pOZ176 in *P. aeruginosa* PA96, was identified as IncP-2 using incompatibility testing methods[19]. Our genomics analysis revealed the presence of a conserved replication and stability system in the core backbone of the megaplasmid family (Supplementary Data 2). The RepA proteins share from 92% to 100% sequence identity (Supplementary Fig. 8) suggesting that all the members of the family belong to the same incompatibility group.

IncP-2 plasmids have been studied for many years, are considered ubiquitous in the environment and are associated with tellurite resistance[20]. We identified tellurite resistance genes (*terABCDEFZ*) as part of the megaplasmid family core genome (Supplementary Data 2).

**Phylogenetic analysis of the pBT2436-like megaplasmid family**. The pangenome analysis identified several AMR genes occurring in multiple copies (from 2 to 5) in members of the megaplasmid family (Supplementary Fig. 6). For example, the *P. putida* plasmid pSY153-MDR features four copies of the gene encoding the multidrug transporter EmrE, whereas pOZ176 harbours three copies of the gene *qacA*, which is absent from the other megaplasmids and is involved in resistance against antiseptic and disinfectant compounds. The efflux pump genes detected in the RR2 of pBT2436 were also present in six other members of the megaplasmid family (Fig. 2). However, there was variation in flanking genes between the megaplasmids.

We constructed phylogenetic trees based on (a) the nucleotide composition of a set of curated core genes (*n* = 105) and (b) the pangenome (presence or absence of protein clusters) (Fig. 4).

Using both approaches, we identified clusters of (i) five megaplasmids isolated in China (including one found in *P. putida*), (ii) two megaplasmids pOZ176 (from China) and pJB37 (from Portugal[21]), and (iii) two megaplasmids for which we have very little information (from strains AR_0356 and AR441). The two other megaplasmids from non-*aeruginosa* hosts and that from the *P. aeruginosa* strain AR439, did not cluster with any other plasmids, or each other. Interestingly, the two megaplasmids from the same hospital reported in this study (pBT2436 and pBT2101) cluster more closely by pangenome analysis than they do by core genome analysis, suggesting some convergence due to acquisition of accessory genes, which, in these plasmids, are mostly AMR related (Fig. 4). The clustering seen in the phylogenetic trees was broadly reflected in the grouping obtained according to AMR gene content (Fig. 5). Two of the megaplasmids found in non-*aeruginosa* strains (p1 and pRBL16), both from non-human sources, lacked any AMR genes, whereas plasmid pST153-MDR, isolated from a *P. putida* associated with a urinary tract infection in China, shares a similar AMR gene profile to other *P. aeruginosa* isolates from China (Fig. 5).

**Wider distribution of the pBT2436-like megaplasmid family**. To determine the wider distribution of the megaplasmid family, we targeted information available in public databases using a strategy of aligning nucleotide sequence data from genomes assembled to different levels but mostly fragmented and produced by short-read sequencing technologies to the complete pBT2436 nucleotide sequence.

We first targeted a data set of 390 genome sequences of *P. aeruginosa* clinical isolates from various infection and geographical sources and including many isolates associated with AMR[22], detecting 10 genomes (~2.6%) displaying estimated pBT2436 coverage values ranging from 21% to 95%

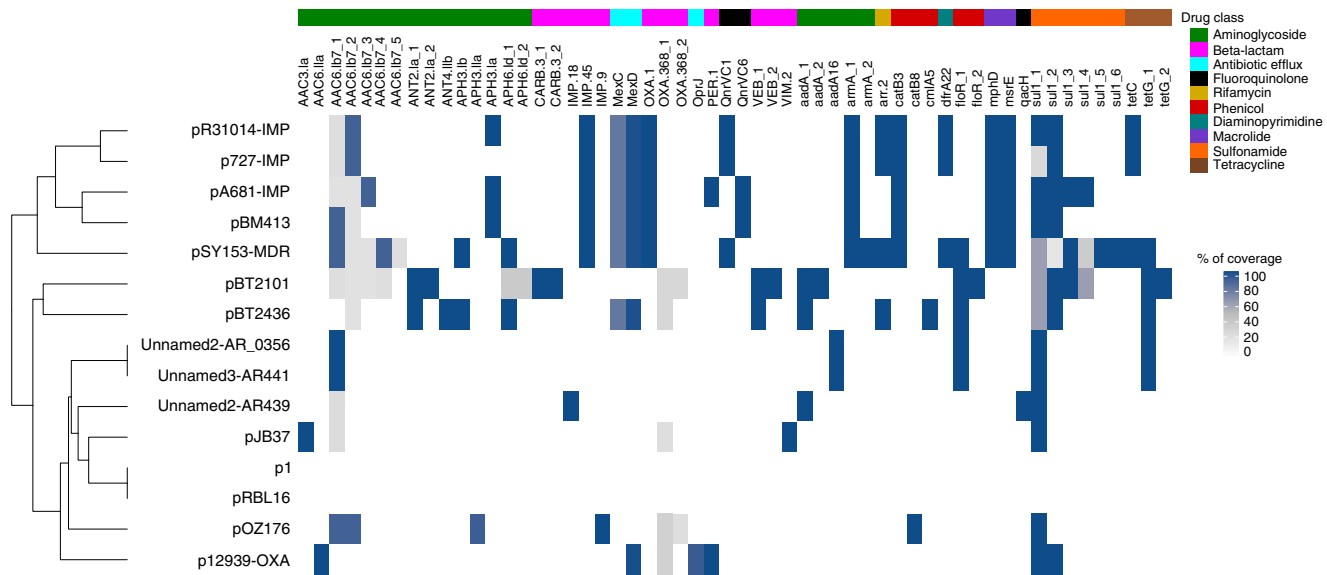

**Fig. 5 AMR gene content of the pBT2436-like megaplasmid group.** The heatmap shows the collection of AMR genes identified in 15 members of the pBT2436-like megaplasmid family through BLASTn searches against the Comprehensive Antibiotic Resistance Database (CARD)[56]. Percentage of coverage of the AMR genes identified from the searches is indicated. The megaplasmids (Y axis) are hierarchically clustered based on their content of AMR genes (X axis) using the "complete" method with Euclidean distance[57]. The "_N" suffix in gene names denotes cases where the gene occurs more than once in the same megaplasmid. AMR genes are additionally classified based on the drug class they confer resistance to according to CARD[56].

(Supplementary Fig. 9; Supplementary Data 4), with 8 genomes having coverage values >75%. These included genomes from isolates obtained from respiratory tract infections (4), intra-abdominal infections (2) and urinary tract infections (2), isolated in countries from Asia (China, India), the Americas (Mexico, Argentina and the United States) and Europe (Germany). All eight isolates were assigned to different MLST groups[22] (Supplementary Data 4), and the oldest was obtained in 2005.

We next targeted a wider database, searching for pBT2436-related megaplasmids in all the genomes from the *Pseudomonas* genus deposited in GenBank. At the time of our search, >5000 genomes were available at GenBank under different assembly categories for both *aeruginosa* (complete: 127; chromosome: 27; scaffold: 1053; contig: 1782) and *non-aeruginosa* (complete: 219; chromosome: 89; scaffold: 1266; contig: 1091) *Pseudomonas* species. Seventy-two matches (~1.3%), 62 of them displaying >60% coverage values, were identified in this larger data set (Supplementary Data 4), most from fragmented genomes (contig:33; scaffold:20). Metadata associated with the matching genomes were extracted from their biosample records (Supplementary Data 4). pBT2436-like megaplasmids were associated with strains from diverse geographical origins (The Americas, Asia, Europe and Africa) and isolation sources, including both environmental and from a broad range of infection types (Fig. 6; Supplementary Data 4). The best match, covering 96.6% of the pBT2436 sequence, was to a plasmid present in a strain of *P. montelii* isolated in Japan in 1986. The match with the oldest recorded collection date corresponded to a 1970 *Pseudomonas* sp. isolate from Japan that displayed coverage of 86.3% (Fig. 6). Although most of the matches were identified in *P. aeruginosa* genomes, there were several examples of pBT2436-like megaplasmids found in other species of *Pseudomonas* (Fig. 6; Supplementary Data 4).

MLST profiles obtained from the megaplasmid-carrier genomes portrayed a highly diverse population featuring 33 different STs among the 56 *P. aeruginosa* genomes assigned with a particular ST, thus indicating extensive horizontal gene transfer mediated by the plasmids (Supplementary Data 4 and Supplementary Fig. 10).

A phylogeny of the wider pBT2436-like megaplasmid population, inferred from selected core gene sequences, revealed novel patterns of diversity previously unrecognised with the comparison of complete plasmids only (Figs. 4 and 6). Although the overall topology of the two phylogenetic trees is similar, we identified new clusters entirely formed by sequences recovered from our megaplasmid search in *Pseudomonas* genomes from GenBank (Fig. 6). The most abundant groups were represented by the plasmids p1 and pOZ176 but several other sub-clusters and an apparent outlier were distinguished as well. Notably, only 4 phylogenetic markers were required to infer the tree suggesting that these genes could form the basis of a typing system.

**Megaplasmids stability and fitness costs.** The stability of pBT2436 was tested by cycling the host bacterium through three rounds of growth in non-selective media (approximately 60 generations) and screening for the maintenance of tobramycin resistance. We detected no loss of resistance and confirmed plasmid maintenance using PCR assays.

We then investigated the conjugative transfer capability of the pBT2436-like megaplasmids, selecting p1 from *P. koreensis* P19E3 and pOZ176 from *P. aeruginosa* PA96 as representatives of the group (see "Methods"). Both p1 and pOZ176 transmitted readily to *Pseudomonas fluorescens* SBW25, demonstrating that the conjugative machinery of pBT2436-like plasmids is functional (Supplementary Fig. 11; see "Methods"). To understand the fitness consequences of megaplasmid acquisition, we performed competitive fitness assays, in which megaplasmid-bearing *P. fluorescens* directly competed against isogenic, differently marked, plasmid-free competitors[23]. Surprisingly, despite their large size, we found that neither pOZ176 nor p1 levied a significant cost on growth in competition (Fig. 7a). Fitness benefits of plasmid carriage in the absence of antibiotics have been previously reported[24]. We detected considerable transmission of both plasmids into the competitors, with between 23% and 55% of competitors having acquired pOZ176 during the course of the competition (Fig. 7b), and detected no loss of pOZ176 or p1. Consistent with our genome analyses, these experimental data

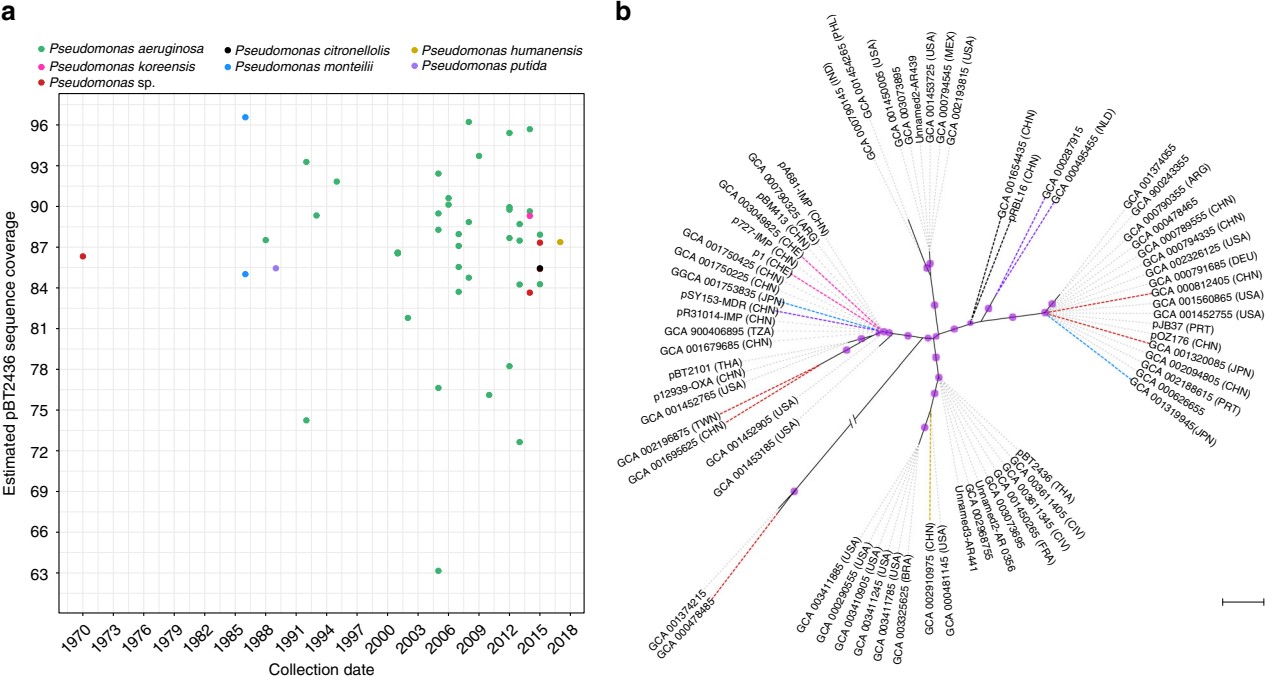

**Fig. 6 pBT2436-like megaplasmids recovered from *Pseudomonas* genome assemblies deposited in GenBank.** Nucleotide sequences of *Pseudomonas* genomes from the four GenBank assembly categories were aligned to pBT2436 to identify unreported or overlooked related megaplasmids. Search was performed in November 2018. **a** The chart shows the pBT2436 estimated coverage and collection date reported in the metadata (when available) of matches covering >60% of the pBT2436 sequence (Supplementary Data 4). Bacterial species of the matches are colour coded and indicated in the figure. **b** Phylogeny of pBT2436-like megaplasmids recovered from *Pseudomonas* assemblies. The maximum-likelihood unrooted tree (GTR+F+ASC+R2 model) was inferred from the alignment of 4 core genes selected as phylogenetic markers (see "Methods") and represents the relationships among 53 sequences from GenBank displaying >80% pBT2436 coverage and 15 complete megaplasmids (bold letters). Dotted lines connect branch tips with the corresponding taxa names and sources (three-letter codes in parentheses, where known). Lines corresponding to non-aeruginosa species' sequences are colour coded as in **a**. Purple dots indicate posterior probability values >0.7. The scale bar corresponds to 0.1 expected substitutions per site. No clear correlations between cluster composition and geographical origin or bacterial species of its taxa are observed.

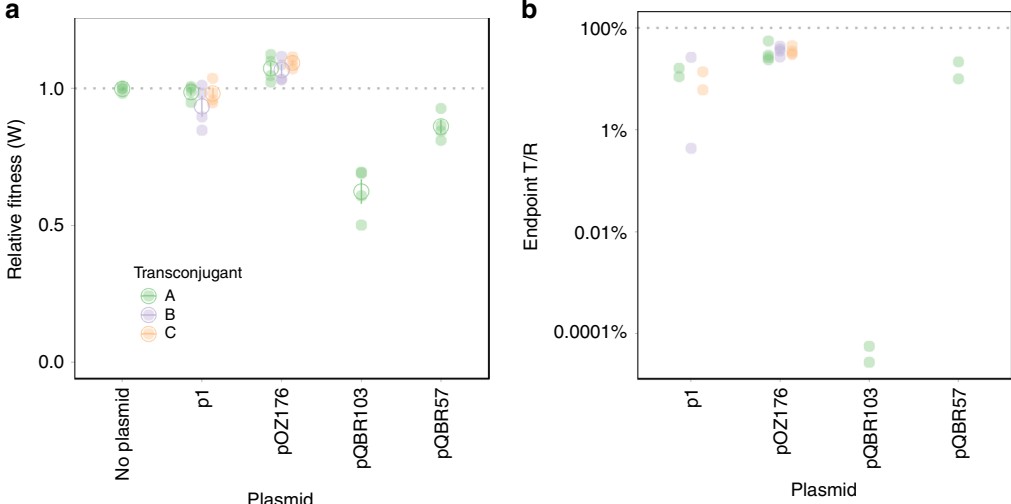

**Fig. 7 pBT2436-like megaplasmid conjugation and fitness cost. a** The relative fitness (growth in competition with plasmid-free) of *P. fluorescens* SBW25 with one of two different pBT2436-like megaplasmids was measured from quadruplicate experiments for three independent transconjugants. Filled points show separate measurements; the open point and bars indicate means calculated for each transconjugant and standard error. pQBR103 and pQBR57 (one transconjugant each), known to be costly, are included as a positive control for plasmid cost. We detected a significant, positive effect of pOZ176 acquisition on fitness (Tukey's HSD pOZ176 vs no-plasmid comparison padj = 0.009) and no significant effect of p1 acquisition (padj = 0.44). **b** Plotted are the ratios of transconjugants (T) to recipients (R) from the endpoints of the competition assays (see "Methods"). Note that we retrieved transconjugant data from only 2/4 experiments with the mercury-resistant plasmids (p1, pQBR103, pQBR57). Source data is provided as a Source Data file.

indicate that pBT2436-like megaplasmids could be a powerful vehicle for the spread of resistance among Pseudomonads in the environment, due to low fitness costs, high rate of transmission and extensive gene content.

## Discussion

Surveillance using whole-genome sequencing approaches has entered the public health arena[25]. Here we show the importance of generating complete genome data using a technology based on longer sequence reads. This enabled us not only to identify a family of MDR megaplasmids associated with the spread of resistance in a hospital in Thailand but also to identify subtle recombination and duplication events, indicative of mosaic AMR regions and dynamic evolution. The association of key adaptive traits with duplicated regions has also been reported for the p1 megaplasmid-carrying *P. koreensis* strain P19E3[17], with genes encoding heavy metal resistance, aromatic compound degradation and DNA repair occurring within large repeats. Consistent with our observations, repetitive genes in p1 were identified close to transposase and phage genes, suggesting a key role for mobile elements in shaping these large genome duplications[17].

The megaplasmids pBT2436 and pBT2101 carry a diverse array of AMR genes. In comparison to pBT2101, pBT2436 carries two additional aminoglycoside resistance genes (APH(3″)-Ib, ANT (4′)-IIb), both of which have been previously linked to amikacin resistance[26,27], the only antibiotic susceptibility difference we detected between the isolates 2436 and 2101. Various mechanisms can contribute to amikacin resistance[28], including efflux pumps. Hence, it is also possible that the additional MexCD-OprJ efflux pump of pBT2436 contributes to amikacin resistance, though previous studies have suggested that MexCD-OprJ plays a lesser role than other efflux pumps in aminoglycoside resistance[29]. Although genes encoding MexCD-OprJ were also found in the chromosome of the carrier strain 2436, the *mexCD-oprJ* region on plasmid pBT2436 displayed higher sequence similarity to chromosomes and plasmids from different organisms than to the chromosome of 2436, thus highlighting the capability of this megaplasmid family to collect and transport genetic information across a wide phylogenetic distance.

We characterised the distribution of multiple repeated regions within and between the two plasmids pBT2436 and pBT2101 and identified the shuffling of genes, revealing high levels of dynamism in the megaplasmid AMR gene regions. It has been reported that *mer* determinants in bacteria are commonly found on plasmids as part of transposons and display a wide variety of arrangements[30]. Consistent with this observation, the *mer* operons identified both in pBT2436 and pBT2101 were located next to transposases or integrases, although of different classes, implying distinct origins.

Our analysis of a small collection of clinical isolates from the same hospital in 2013 revealed four isolates carrying related megaplasmids, including two obtained from the same patient but from different clinical samples, and two from different patients but present in isolates sharing nearly identical MLST (isolates 2101 and 3583). The latter is consistent with a cross infection or common source event, but it is notable that the plasmids have diverged with respect to a large duplication of AMR genes during the event. The presence of MDR (including carbapenem resistance) megaplasmids in 4 of only 23 isolates analysed (~17.4%) suggests that the megaplasmids may be playing an important role in the spread of resistance in this clinical setting and that a more contemporary wider surveillance study is urgently needed.

After further exploration of the available repository of complete genomes, we identified several related plasmids, mostly harboured by strains from China, consistent with the notion of a family of carrier megaplasmids acting as vehicles for the spread of AMR genes. These included pBM413, a MDR megaplasmid[14,31] carrying *qnrVC6* and *bla*$_{IMP-45}$ in *P. aeruginosa* strain Guangzhou-Pae617[32], a sputum isolate from a patient suffering from respiratory disease, and pOZ176, containing two integrons harbouring *bla*$_{IMP-9}$ and *bla*$_{OXA-10}$, respectively, and isolated from a MDR strain obtained from the sputum of an intensive care unit patient (PA96), following an outbreak of carbapenem-resistant *P. aeruginosa*[19,33]. The MDR megaplasmid pSY153-MDR, carrying *bla*$_{IMP-45}$, was discovered in a *P. putida* isolate from the urine of a cerebral infarction patient in China[14]. Despite being present in a different species, this plasmid is more closely related to the other plasmids isolated from China than to different members of the megaplasmid family, suggesting that between-species transmission occurred locally. The isolation of a divergent MDR megaplasmid from a *Pseudomonas shirazica* clinical isolate further supports this notion[34].

There were extensive variations in the carriage of AMR genes among members of the megaplasmid family, including two megaplasmids lacking AMR genes altogether. pRB16 was identified in *P. citronellolis* SJTE-3, an oestrogen and polycyclic aromatic hydrocarbon degrading bacterium isolated from active sludge at a wastewater treatment plant in China[35]. *P. koreensis* P19E3, harbouring the megaplasmid p1, was isolated from healthy marjoram (*Origanum majorana*) leaf material in an organic herb farm (Boppelsen, Switzerland) in 2014[17]. The presence on p1 of a cluster of genes encoding copper resistance suggests that, while members of the megaplasmid family share a common backbone, they are flexible in the adaptive traits that they carry.

By mining databases dominated by contigs assembled from short-read data, we gained valuable insights into the ecology and evolution of the megaplasmid family. Our findings indicated that members of the megaplasmid family are widely distributed geographically and have been isolated from various sources, both environmental (including sewage, sludge, high-arsenic soil, lake water, plant material and phenol treatment bioreactors) and clinical (including sputum, pneumonia, urine, faeces, blood, keratitis, cerebrospinal fluid and burn samples). Recent characterisation of *P. aeruginosa* industrial isolates carrying related megaplasmids further supports our observations on the wide distribution of the pBT2436-like family[36]. We were able to detect plasmid sequences highly similar to pBT2436 in genomes from at least six different *Pseudomonas* species. Several matches from non-*aeruginosa* species displayed higher coverage values than others detected from *P. aeruginosa* isolates, suggesting active and recent megaplasmid transfer within the genus. This idea is consistent with the core and pangenome trees constructed from complete megaplasmid sequences, where we observed a clear clustering of the *P. putida* megaplasmid pSY153-MDR with other members of the family hosted by *P. aeruginosa* isolates. The serotype and MLST metadata available for some of the analysed genomes confirm that the pBT2436-like megaplasmids are widely distributed within the *P. aeruginosa* population.

By analysing databases that include genome data from isolates obtained decades ago, we were able to obtain insights into the temporal development of the megaplasmid family. Although most of the isolates harbouring related plasmids were isolated after the year 2000, reflecting the fact that the database is dominated by more recent isolates, nine of them were collected prior to the year 1995. The three oldest isolates, dating back to 1970–1986, corresponded to non-*aeruginosa Pseudomonas* species from Japan. It is tempting to speculate on an evolutionary process whereby megaplasmids "empty" of AMR genes assembled in non-*aeruginosa* species, some collecting diverse cassettes of AMR genes either before or after moving into *P. aeruginosa*, with subsequent selection for the MDR-associated arrangements now seen in a

clinical setting. Although much of the data that we present is consistent with this view, the best database match of all to our recent MDR megaplasmid pBT2436 (96.6% coverage) was to a plasmid present in a strain of *P. monteilii* isolated in Japan in 1986, suggesting that the reality is more complex, and that some plasmid/AMR arrangements that we might assume to be recently assembled have existed for some time. It is worth noting that *P. monteilii* has generally been associated with human pulmonary infections[37,38] and the 1986 isolate was from a human source.

Further clues as to the origin and evolution of this megaplasmid family can be found through comparisons with more distant matches. Two environmental mercury resistance plasmids, pQBR103 and pQBR57, have similar gene order and content among backbone genes[23], including the presence of a radical SAM gene cluster, chemotaxis genes and a type IV pilus, though they are both highly divergent at the nucleic acid level and thus were not detected in our survey. The conservation of this long region of synteny between distantly related plasmids suggests a functional association between the genes involved, but exactly what this is remains unclear. Neither pQBR57 nor pQBR103 contain any identifiable AMR genes. They were isolated from pristine agricultural fields, consistent with the hypothesis of a pool of environmental megaplasmids, some of which have adapted to become vectors of clinically relevant AMR.

Plasmid carriage can impose fitness costs to host bacteria through the burdens associated with acquisition of superfluous genes, and for these megaplasmids, increasing the number of open reading frames (ORFs) by up to ~9%, these costs might be restrictive. However, discrepancy in size between large and small plasmids is also often balanced by reciprocal variation in copy number such that the total burden in terms of DNA replication is equivalent[39], and the contribution of size per se to plasmid cost can be relatively small compared with costs associated with gene expression[40,41]. Megaplasmids may, therefore, be no less effective at AMR maintenance and transmission than smaller vectors. Furthermore, the fact that mutations can ameliorate even costly megaplasmids suggests that costs may not inhibit megaplasmid success in the long term[40,42,43]. Our study highlights a widespread megaplasmid family present across clinical, environmental and geographical sources, and in multiple species hosts, suggesting widespread maintenance of this large plasmid backbone and selection for maintenance of this backbone beyond adaptive traits such as AMR. The data provide important insights into how plasmids in the environment can act as a reservoir for the emergence of important traits such as AMR and MDR plasmids in a clinical setting.

## Methods

**Bacterial isolates**. The 48 clinical isolates used in this study were acquired in 2013 from patients in Ramathibodi Hospital, Mahidol University, Bangkok, Thailand and were associated with different types of infection (Supplementary Data 1). Antimicrobial susceptibility testing was carried out according to the EUCAST guidelines[44]. Briefly, isolates to be tested were cultured onto Columbia plates (overnight 37 °C). From these, single colonies were mixed with sterile distilled water to attain a standard optical density (0.5 McFarland units), and 10 µl spread was onto Mueller–Hinton agar plates with a swab and incubated overnight at 37 °C with Meropenem (10 µg), Ceftazidime (30 µg), Piperacillin/Tazobactam (75/10 µg), ciprofloxacin (5 µg) or tobramycin (10 µg) antibiotic discs. Antibiotic susceptibility was determined by measuring the zone of growth inhibition, and the results were interpreted as sensitive, intermediate or resistant, according to the EUCAST guidelines[44].

**Whole-genome sequencing**. *Short-read sequencing*: Genomic DNA (500 ng) was extracted from overnight cultures and sequenced using an Illumina MiSeq at the Plateforme d'Analyses Génomiques of the Institut de Biologie Intégrative et des Systèmes (Laval University, Quebec, Canada)[45].
*Long-read sequencing*: Genomic DNA (15 µg) was extracted from overnight cultures using a Promega Wizard Genomic DNA Purification Kit, quantified using a Qubit 3.0 fluorimeter (Qubit dsDNA broad range assay kit, Life Technologies)

and tested for purity using a NanoDrop 1000 spectrophotometer (Thermo Scientific). Single-molecule real-time (SMRT) sequencing was performed at the Centre for Genomic Research, University of Liverpool. For each genome, three SMRT (RSII SMRT) cells with P6/C4 chemistry on a Pacific Biosciences (PacBio) were used to generate the raw sequence data.

**Genome assembly**. PacBio reads were de novo assembled using the HGAP v.3 workflow[46] through the Pacific Biosciences SMRT Portal v.2.3.0. The RS_HGA-P.3_Assembly.3 protocol was run with default settings, except that the estimated genome size was set to 6.5 Mb. Contigs featuring either <20× coverage, <20 kb length, <47 Quiver-based quality values (QV) or >95% sequence identity to larger contigs with better coverage and QV were discarded from the assembly. Surviving contigs were assessed for circularisation with Circlator v.1.5.3[47], followed by further polishing with both the RS_Resequencing.1 protocol (SMRT Portal) and PILON v.1.22[48] using the corresponding Illumina sequencing reads. Genes *dnaA* and *repA* were set as the start position of the chromosomes and plasmids, respectively. PacBio genome assembly and genome annotation were performed using a virtual machine hosted by the Cloud Infrastructure for Microbial Bioinformatics consortium[49].

For Illumina sequencing, raw Fastq files were processed for adapters and quality trimming using Cutadapt v.1.2.1[50] and Sickle v.1.200 (https://github.com/najoshi/sickle). The genomes were de novo assembled and scaffolded following the A5 MiSeq assembly pipeline[51] as described in ref. [52]. Genome assemblies' quality was assessed with QUAST v.3.1[53].

**Annotation**. The MLST profiles of the genomes were identified from the pubMLST *P. aeruginosa* scheme (http://pubmlst.org/paeruginosa/)[54] using the mlst tool v.2.8 (https://github.com/tseemann/mlst). AMR genes were detected through BLASTn searches[55] against the Comprehensive Antibiotic Resistance Database[56] using the ABRicate tool v.0.8 (https://github.com/tseemann/abricate). Only matches displaying coverage >20% were selected to generate the heatmap representing the AMR gene content of the megaplasmids group with the ComplexHeatmap package v.1.17.1[57]. Genome annotation was carried out with Prokka v.1.12[58] for both the sequences reported in this study and the complete megaplasmids identified in GenBank (Table 1) to homogenise the annotation process. All the sequences were permuted to set the *repA* gene as the start position prior to annotation. pBT2436/pBT2101 genome annotation was complemented with the Sma3s tool v.2[59] and by integrating data from conserved domain searches performed with InterProScan v.5.30-69.0[60]. Integrons in the Thai megaplasmids were identified using the Integron Finder package v.1.5.1[61]. ISs in the plasmid resistance regions were detected and annotated through the BLASTn tool of ISfinder[62].

**Comparative analysis**. Pairwise comparison of the pBT2436 and pBT2101 megaplasmid sequences was performed through BLASTn[55] and visualised with the Artemis Comparison Tool (ACT) v.17.0.1[63]. Repeated sequences in the pBT2436 and pBT2101 resistance regions were identified by aligning the regions against themselves with BLASTn[55] and visualising the output with ACT[63], masking the largest match corresponding to the overall sequence self-identity. Sequences homologous to the pBT2436 resistance region 2 encoding the MexC-MexD-OprJ efflux pump were detected through BLASTn[55] searches against the non-redundant (Nr) GenBank database and the pairwise alignments visualised with ACT v.17.0.1[63].

Complete pBT2436-like megaplasmid sequences deposited in GenBank were found through BLAST searches[55] against the Nr nucleotide collection using pBT2436 as query sequence. The search was performed in November 2018 and only hits featuring <1e−5 E-value, >75% Query coverage, and >1e5 Max score were selected as complete homologous sequences. Note that all the non-selected sequences displayed query coverage values <25%. The circular representation of the genome comparison between the complete megaplasmids and pBT2436 or pBT2101 was generated using the BRIG application v.0.95[55,64].

In this paper, we refer to pangenome analysis only in the context of the megaplasmid family and not the chromosome. The pangenome analysis of the pBT2436-like megaplasmid group was performed with the GET_HOMOLOGUES package v.10092018[16,65–68]. The GET_HOMOLOGUES pipeline was run with default settings except for the additional "-E 1e-03 -c -z" options. Core and Pangenome were computed with the compare_clusters.pl script as the intersection of the BDBH-COG-OMLC and COG-OMCL algorithms, respectively. Rarefaction curves were generated using the plot_pancore_matrix.pl script. The pangenome matrix obtained from the analysis was visualised as a heatmap with the ComplexHeatmap package v.1.17.1[57]. Functional annotation of the accessory proteins of the group was performed using Sma3s v.2[59].

The phylogenomic analysis of the megaplasmids group was carried out following the GET_PHYLOMARKERS pipeline v.2.2.5[69–72]. The set of single-copy core-genome clusters detected with GET_HOMOLOGUES was input into the pipeline to identify high-quality phylogenetic markers and inferring a core genome phylogeny through a maximum-likelihood tree search. The pipeline was run under the following settings: phylogenetics mode, DNA sequences, IQ-TREE as the tree-searching algorithm, "high" model evaluation mode for IQ-TREE (based on ModelFinder) during the gene-trees searching, and 10 independent IQ-TREE tree

searches on the concatenated top-scoring markers ("-R1 -t DNA -A I -T high -N 10" options). A pangenome phylogeny was inferred from the gene content profiles of the members of the group using the estimate_pangenome_phylogenies.sh script. A maximum-likelihood tree search was performed on the pan-genome matrix built from the GET_HOMOLOGUES analysis using the binary and morphological models implemented in IQ-TREE and launching ten independent tree searches. The phylogenetic trees were visualised and edited with the iTOL tool v.4.3.3[73].

**Search for megaplasmid homologues and sequence comparison.** The corrected Illumina-generated sequencing reads from the Thai isolates were mapped to pBT2436 searching for related megaplasmids using BWA-MEM v.0.7.17-r1188[74]. The presence of megaplasmids was assessed from the percentage of reads mapped to pBT2436 as extracted from the alignment files with samtools v.1.7[75]. Genomes displaying <1% of mapped reads were considered as lacking pBT2436-like mega-plasmids based on the value obtained for the isolate 4068 genome used as negative control. Coverage of the pBT2436 sequence was assessed and visualised with BRIG v.0.95[64]. Similarity among the Thai isolate genomes was estimated through kmer-based sequence clustering using the SaffronTree v.0.1.2 algorithm[76].

The search of megaplasmids homologous to pBT2436 in the larger database was performed with NUCmer from the MUMmer package v.3.1[77]. The query data set was obtained from the GenBank entries reported in ref. [22] and the collection of *Pseudomonas* genomes available at the GenBank assembly database (https://www.ncbi.nlm.nih.gov/assembly) in November 2018. The query sequences were aligned to pBT2436 running nucmer with default settings but using anchor matches that were unique to both the reference and query sequences (-mum). pBT2436 percentage of coverage was estimated by adding up the reference coverage values of all the alignments listed in the .coords files generated with the show-coords utility. Plots displaying the pBT2436 coverage, percentage of sequence identity and location of the aligned regions were generated using the mummerplot script. Metadata associated with queries matching the pBT2436 sequence were extracted from the corresponding biosample records. The MLST profile of the genomes identified as megaplasmid carriers was obtained with the mlst tool v.2.8 (https://github.com/tseemann/mlst) from the closest *Pseudomonas* scheme available (https://pubmlst.org/databases/).

The broader diversity of the pBT2436-like plasmid family was explored by comparing the 15 complete megaplasmid genomes to sequences identified in GenBank as megaplasmid carriers. We based our strategy in the detection of core megaplasmid genes, hence we restricted the comparison to genomes from GenBank displaying at least 80% of the pBT2436 sequence in the homologous search ($n = 53$). The set of core genes previously detected from the pangenome analysis of the complete megaplasmids was first compared to the *P. aeruginosa* PAO1 genome (accession: NC_002516) through BLASTn[55] searches to ensure that these were plasmid specific. The selected *Pseudomonas* genomes were annotated with Prokka v.1.12[58] to homogenise the annotation prior the comparative analysis (several genomes had no annotation available in GenBank). Both nucleotide and protein sequences of the 348,945 ORFs detected in the previous step were compared to our set of 261 core megaplasmid sequences using GET_HOMOLOGUES v.10092018[16] (default settings plus "-E 1e-03 -t 68 -e -c" options) to recover megaplasmid genes from the *Pseudomonas* genomes. Only core plasmid genes present in all the samples ($n = 104$) were selected for the downstream analysis. Selection of markers among the recovered core genes and construction of the phylogenetic tree were performed with GET_PHYLOMARKERS v.2.2.5[69]. A maximum-likelihood phylogeny was inferred from the concatenated sequences of the top four markers (pBT2436_00034 (*cheZ*), pBT2436_00070 (hypothetical protein), pBT2436_00133 (*yceC*), pBT2436_00135 (*terB*)).

**Plasmid stability and fitness cost.** Plasmid stability was determined by culturing isolate 2436 through 3 rounds of overnight growth in Lysogeny broth at 37 °C (approximately 60 generations) and screening 100 colonies for maintenance of tobramycin resistance (256 µg/ml). All the colonies maintained resistance. Plasmid maintenance was then confirmed for 20 colonies using PCR assays for 3 mega-plasmid core genes (*repA*, *parA* and *virB4*, Supplementary Table 2).

The widespread presence of pBT2436-like megaplasmids across genera, and their carriage of genes encoding conjugative pili, suggests that they are capable of conjugative transfer. The broad resistance profile conferred by pBT2436 and pBT2101 complicates transmission assays, as selecting, enumerating and isolating transconjugants usually requires availability of a selectable marker that is carried by the recipient and not the plasmid. Therefore, we carried out transmission assays on two other representative members of this family, namely p1 from strain P19E3[17] and pOZ176 from strain PA96[19]. These two plasmids have >99.5% backbone gene average nucleotide identity with pBT2101 and pBT2436, but neither encodes streptomycin resistance, enabling selection of transconjugants by testing for mobilisation of mercury resistance (encoded by p1) or kanamycin resistance (encoded by pOZ176) into a streptomycin-resistant recipient. To understand the ability of MDR megaplasmids to be maintained and spread among bacteria in the environment, as a recipient we used the model soil pseudomonad *P. fluorescens* SBW25, which has the additional advantage of having previously been used to understand megaplasmid evolution and ecology[42].

To transfer pBT2436-like megaplasmids into *P. fluorescens* SBW25, equal volumes of overnight cultures of donors (*P. koreensis* P19E3[17] or *P. aeruginosa* PA96[17,19]) and recipients (*P. fluorescens* SBW25 labelled with lacZ and streptomycin resistance, henceforth *P. fluorescens* SBW25-SmR-lacZ[23]) were mixed, and 60 µl was used to inoculate 6 ml King's B (KB) broth without antibiotics. KB consists of 20 g/l Bacto Proteose Peptone #3, 1.5 g/l MgSO$_4$•7H$_2$O, 1.5 g/l K$_2$HPO$_4$•3H$_2$O and 10 g/l glycerol. Solid KB included 12 g/l agar and 50 µg/ml X-gal to enable distinction of *P. fluorescens* SBW25-SmR-lacZ. Six independent cultures were established for each plasmid. After 48 h growth at 28 °C 180 rpm shaking, samples were plated onto KB agar supplemented with 250 µg/ml streptomycin and 50 µg/ml X-gal and either 50 µg/ml kanamycin (for pOZ176 conjugations) or 20 µM mercuric chloride (for p1 conjugations) and incubated at 28 °C for 48 h. All colonies that grew resembled plasmid-free *P. fluorescens* SBW25-SmR-lacZ, producing blue pigment due to carriage of lacZ. Putative transconjugants were restreaked on selective media to ensure clonality before stocking. Plasmid acquisition was confirmed by PCR (Supplementary Table 2). Colony PCR assays were performed using New England Biolabs OneTaq with 0.4 µM each primer and the following cycling conditions: 94 °C 5′, 30 × (94 °C 30″, 55 °C 30″, 68 °C 1′), 68 °C 5′. Similar colony PCRs targeting genes located on other *P. koreensis* P19E3 MGE (three other plasmids, two of which are mobile, as well as chromosomal conjugative machinery) yielded no products from the *P. fluorescens* transconjugant strains, suggesting that only the megaplasmid of interest had transferred.

Competition assays were established from overnight cultures of *P. fluorescens* SBW25-SmR-lacZ (test strain, produces blue colonies on KB supplemented with X-gal) and *P. fluorescens* SBW25 with gentamicin resistance (henceforth *P. fluorescens* SBW25-GmR, reference strain, produces white colonies). Test strains were plasmid free with the pBT2436-like megaplasmids pOZ176 or p1 or with previously characterised plasmids pQBR57 or pQBR103, which are known to confer fitness costs of ~15% and ~50%, respectively[23,42]. Three independent transconjugant strains were tested for pOZ176 and p1 to control for any stochastic effects occurring during strain generation; we used only one transconjugant for pQBR57 and pQBR103 as these were known to be representative[23]. Four replicate competition assays were conducted for each strain. For each competition, approximately equal numbers of test and reference strains were mixed, and samples were spread onto KB agar with 50 µg/ml X-gal for enumeration of starting population. Mixtures were diluted 100-fold into 6 ml KB broth and incubated at 28 °C 180 rpm shaking for 48 h before samples were spread for endpoint counts as described above. Relative fitness ($W$) was calculated as the ratio of Malthusian parameters $W = (\ln(\text{testend/teststart}))/(\ln(\text{referenceend/referencestart}))$[78]. We detected no significant effect between transconjugants for either plasmid (Kruskal–Wallis Test pOZ176 $\chi^2 = 0.5$, $p = 0.78$; p1 $\chi^2 = 1.19$, $p = 0.55$), so we analysed all transconjugants of pOZ176 and p1 together in comparison with plasmid free using analysis of variance.

To enumerate mercury-resistant transconjugants (i.e. for p1, pQBR57 and pQBR103), samples of endpoint cultures were spread on KB agar supplemented with 30 µg/ml gentamicin, 20 µM mercuric chloride and 50 µg/ml X-gal. Transconjugants were identified as white colonies (no blue colonies were detected). Loss of p1 was assessed by patching ≥12 blue colonies from endpoint plates onto KB + 100 µM mercuric chloride; no loss was detected. To enumerate kanamycin-resistant transconjugants (i.e. for pOZ176), endpoint plates were replicated onto KB agar containing 50 µg/ml kanamycin and 50 µg/ml X-gal. Transconjugants were identified as white colonies that transferred, and loss of pOZ176 was assessed by screening for blue colonies that did not transfer. No loss of pOZ176 was detected.

**Reporting summary.** Further information on research design is available in the Nature Research Reporting Summary linked to this article.

## Data availability

Accession numbers to the Illumina-sequenced genomes from the Thai isolates reported in this work, and their corresponding sequencing reads, are listed in Supplementary Data 1. The genome sequences and raw sequencing reads of the 2101, 2436 and 4068 isolates generated with the PacBio platform have been filed under the GenBank BioProject PRJNA540594. The source data underlying Fig. 1 and Supplementary Fig. 11 are provided as a Source Data file.

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

## Acknowledgements
We thank Michael A. Brockhurst (University of Sheffield, UK) for his contributions and critical feedback after kindly reviewing this manuscript. We also thank Paul Roy (Université Laval, Canada) via Edze Westra (University of Exeter, UK) for the gift of *P. aeruginosa* PA96 and Mitja Remus-Emsermann (University of Canterbury, New Zealand) for the gift of *P. koreensis* P19E3. This work was partially supported by the International Pseudomonas Genomics Consortium, funded by Cystic Fibrosis Canada [to R.C.L.]; the Secretaría de Educación, Ciencia, Tecnología e Innovación (SECTEI), Mexico [to A.C.; Grant seciti/069/2017]; the Medical Research Foundation [to J.L.F. and L.L.W.; Grant MRF-091-0006-RG-FOTHE] and through the Institutional Strategic Support Fund (ISSF) awarded by Wellcome Trust via the University of Liverpool [to J.P.J.H.; Grant 204822/Z/16/Z].

## Author contributions
A.C. and C.W. conceptualised, designed the study and wrote the paper with input from the other authors. A.C. carried out the bioinformatics analyses, data curation and visualisation. J.P.J.H. designed the PCR primers and performed the conjugation and fitness assays. C.W. and J.L.F. supervised the work and reviewed the manuscript. J.L.F. participated in the study design. M.P.M., J.-G.E.-R. and R.C.L. generated and assembled the Illumina-sequenced genomes. M.G. and L.L.W. performed the antimicrobial susceptibility testing. L.L.W. also contributed in performing the pBT2436 stability test. P.P. and P.S. collected the clinical isolates and the associated metadata.

## Competing interests
The authors declare no competing interests.
