## [Peer Review File · Nature Communications]

Reviewers' comments:

Reviewer #1 (Remarks to the Author):

In this paper, the authors sequenced two new closely related megaplasms that carry a variety of AMR genes from isolates of *Pseudomonas aeruginosa* collected in a hospital in Thailand. These plasmids contain complex resistance regions that shown evidence of gene acquisition and duplication, highlighting the dynamic evolution of AMR-associated regions of the plasmids. The authors then shown that this family of megaplasms is found in a diversity of *Pseudomonas* isolates, including species other *P.aeruginosa*.

The authors are correct in stating that the role of plasmids in the spread of AMR in *P.aeruginosa* has been largely overlooked, and genomic and evolutionary analysis presented in this paper goes well beyond most papers describing new AMR plasmids in *Pseudomonas*.

Although the comparative genomic analysis presented in this paper is clear and careful, I think that the paper is overly descriptive, and I think that this study would probably be better suited for a more specialized journal. Although it is clear that these plasmids have been overlooked, it is unclear why these megaplasms play an important role in resistance evolution. For example, I think that it would add a lot to the paper if the authors were able to show that these plasmids have a high stability/impose a low fitness cost, or if the the authors were able to put togethe a convincing model to explain how these plasmids acquired resistance genes (ie what were the sources of resistance).

I have several minor comments that the authors may wish to consider:

-It is interesting that the megaplasms have much lower GC (57%) than the *P.aeruginosa* chromosome. I found the supplementary figure detailing GC content difficult to interpret, but AT-rich islands on the plasmids could be used as a marker for recently acquired genes, and it may be possible to infer the evolutionary origin of regions with atypical AT composition.

-A number of approaches have recently been applied to date the divergence of bacterial genomes, such as BEAST and BactDating, and it might be possible to use these approaches to generate a more refined estimate of the date of acquisiotn of multidrug resistance on these megaplasms

Reviewer #2 (Remarks to the Author):

General considerations

In this study, extensive bioinformatic searches have been used to characterize the structure and phylogeny of a number of megaplasms from various geographical, environmental and clinical origins, some plasmids carrying multiple resistance genes to antibiotics, antiseptics and heavy metals. These data conclusively demonstrate that several megaplasms hosted by *Pseudomonas* sp are closely or more distantly related, but belong to a same family, with a possible ancestor dating back to the 1970s or so. The manuscript is well written and scientifically sound.

Being not a specialist of bioinformatics, my comments will relate to other aspects of this paper.

1. If all the plasmids presented here are phylogenetically linked, intuitively one could assume that the most conserved part of their structure would include the genes that are essential for their replication, maintenance and partition. I suggest the authors provide and make a comment on this information.

2. Some of the plasmids presented carry genes of resistance to tellurite. Referring back to the pioneering works on *Pseudomonas* plasmids in the 1970s (e.g. by George Jacoby), large plasmids of incompatibility group IncP2 were considered as associated with tellurite resistance. Actually, nothing is said in the paper about the incompatibility group of this new family of replicons and whether all these contain the *ter* locus.

3. A number of the described plasmids contain widely distributed genes of resistance to different classes of antibiotics. Have attempts to transfer the plasmids to new hosts been attempted to provide further insights into their role in resistance gene diffusion, in correlation with their tra genes content ?

Minor points

4. Abstract and throughout the text : The word "pangenome" is used to describe the whole set of genes present in the considered megaplasmids. I think this is confusing since the notion of pangenome also includes non-plasmidic (chromosomal) genes.

5. A DNA fragment carrying a (nfxB)mexCD-oprJ-like locus has been found in several of these megaplasmids. It is not clear in the text (pages 4 and 7) how this fragment may have been collected from *A. hydrophila* (IS ?). A further comment on this would be useful.

6. Page 4 (bottom), page 5 (top) and Table S3 please replace strain 3582 with strain 3583

7. Page 11. EUCAST does not recommend to spread 10 microliters of a 10 McFarland bacterial suspension for the diffusion method, but rather to use a 0.5 McFarland suspension spread onto a Mueller-Hinton agar plate with a swab. Please, correct.

Reviewer #3 (Remarks to the Author):

The authors report sequencing of two closely related *P. aeruginosa* megaplasmids conferring MDR. Analysis of AMR regions revealed extensive variation, with various duplications and other rearrangements, highlighting the dynamic nature of these regions. Related plasmids were identified by querying publicly available sequence data, demonstrating a wide distribution of the megaplasmid family across geographic locations, source types, and *Pseudomonas* species. The findings provide insight into plasmid-mediated AMR gene dissemination in *Pseudomonas*, and suggest that this megaplasmid family may be an important vector for such spread. The manuscript is well written, thorough, and carefully thought through. I have only a few suggestions for improvement:

1. Page 3: "In each of the three strains we identified circularised plasmid sequences." The text following implies that only a single plasmid was identified in each strain – is this correct? Please clarify wording.

2. Page 3: "...including a 35.9 kb region duplicated but inverted in pBT2101 when compared to pBT2436". I don't understand why pBT2436 is mentioned here. Isn't the duplication simply two copies of the same 35.9 kb sequence in opposite orientations in pBT2101?

3. Page 4: "A further example of how dynamic these regions are can be seen with the repeats 2b and 5b of pBT2436, located next to xerD (Fig. 1). Their pairs occupy different positions in pBT2436 RR1, whereas they occur merged as one repeat separated by 13 bp in the Resistance Region of pBT2101 (Fig. 1)" What exactly is meant by "different positions" in pBT2436? They look to be very close in the figure – how far apart are they exactly?

4. Page 5: The finding that isolates 2101 and 3582, from different patients, have very similar plasmid sequences and almost identical MLST profiles is interesting. From the supplementary table it appears that the MLST profiles have 6/7 exact matches and one partial match – it is tempting to speculate that this partial match may be an assembly artefact rather than a genuine difference. Can the authors provide any additional insight here? Given the possible transmission link, I think it would be worthwhile investigating the relationship between these isolates at a more fine-grained level such as genome-wide SNVs. Also, in the discussion these isolates were described as "sharing the same MLST" – this description should be made consistent with the results section above.

5. Page 8: How genetically diverse were the matching samples from the GenBank genome search?

It would be helpful to include MLST data or similar to get an idea of just how much evidence there is for HGT.

6. Page 11: "increasing the number of open reading frames up to ~9%" => increasing the number of open reading frames **by** up to ~9%??

7. The text in several of the figures is very small and difficult to read. I would suggest revisiting all the figures to ensure that minimum font sizes are maintained.

8. Supplementary Fig. 1: What level of sequence identity is represented in the sequence alignment? Please clarify in the figure legend.

9. Data availability: For the PacBio genome sequences, I cannot access PRJNA540594 - please ensure this is made publicly available. For the Illumina sequences, it appears that some have already been published (Freschi et al 2019. *Genome Biol Evol.* 11(1):109-120). Please clarify which isolates were newly sequenced for this study. In addition, all raw read data (Illumina and PacBio) should be deposited in the SRA.

Response to referees

Re: A megaplasmid family responsible for dissemination of multidrug resistance in *Pseudomonas*

Reviewer #1 (Remarks to the Author):

In this paper, the authors sequenced two new closely related megaplasms that carry a variety of AMR genes from isolates of *Pseudomonas aeruginosa* collected in a hospital in Thailand. These plasmids contain complex resistance regions that shown evidence of gene acquisition and duplication, highlighting the dynamic evolution of AMR-associated regions of the plasmids. The authors then shown that this family of megaplasms is found in a diversity of *Pseudomonas* isolates, including species other *P.aeruginosa*.

The authors are correct in stating that the role of plasmids in the spread of AMR in *P.aeruginosa* has been largely overlooked, and genomic and evolutionary analysis presented in this paper goes well beyond most papers describing new AMR plasmids in *Pseudomonas*.

Although the comparative genomic analysis presented in this paper is clear and careful, I think that the paper is overly descriptive, and I think that this study would probably be better suited for a more specialized journal. Although it is clear that these plasmids have been overlooked, it is unclear why these megaplasms play an important role in resistance evolution. For example, I think that it would add a lot to the paper if the authors were able to show that these plasmids have a high stability/impose a low fitness cost, or if the authors were able to put together a convincing model to explain how these plasmids acquired resistance genes (ie what were the sources of resistance).

- We have now performed a series of experiments showing that the megaplasmid pBT2436 is stable in the absence of antibiotic selective pressure and that resistance is maintained afterwards (tested with Tobramycin). New experiments presented in this revised version of our manuscript also show that pBT2436-like megaplasms from different hosts can be transferred by conjugation to *P. fluorescens*, where they do not impose a fitness cost to the host. We propose that a high transference rate and low fitness cost add to the large gene collection capacity of the megaplasms as factors making them powerful AMR vectors. These findings are described in a new section in Results, entitled “Megaplasms stability and fitness costs”. They are also covered by the new Fig 7, Supplementary Fig. 11, and Supplementary Table 6. Detailed methods corresponding to these experiments are presented in the “Methods” and “Supplementary Methods” sections.
- To gain insights on the acquisition of AMR genes by the megaplasms, we further annotated the pBT2436 and pBT2101 resistance regions searching for Insertion Sequences (IS) (Fig. 1; Supplementary Table 2). These new data show that resistance regions correspond to complex mosaics of transposons and integrons from different bacterial species, although a predominant IS (e.g. TnAs3 from *Aeromonas salmonicida* belonging to the Tn3 IS family) can be recognized. Hence, a variety of IS play a critical role in the acquisition of AMR genes by the pBT2436-like megaplasms from diverse sources. These new findings are presented in the “AMR regions in pBT2436 and pBT2101 are mosaic and dynamic” results section of the revised manuscript:

“We searched for Insertion Sequences (IS) in the megaplasms Resistance Regions to gain insights about the origin of the transposase, integrase, and resistance genes. Matches to eight

different IS were identified scattered across the Resistance Regions, including some within resistance genes, indicating a key role for IS in the acquisition of resistance from different origins. Matches corresponding to Tn5393, IS6100 and TnAs3 were shared by the two megaplasmiids (Fig.1, Supplementary Table 2). Although the origin of the recognized IS is diverse, most of the matches in pBT2436 and pBT2101 corresponded to elements described in *Aeromonas salmonicida*. No matches to known IS were detected in the pBT2436 RR2.”

I have several minor comments that the authors may wish to consider:

-It is interesting that the megaplasmiids have much lower GC (57%) than the *P.aeruginosa* chromosome. I found the supplementary figure detailing GC content difficult to interpret, but AT-rich islands on the plasmids could be used as a marker for recently acquired genes, and it may be possible to infer the evolutionary origin of regions with atypical AT composition.

- Supplementary Fig. 1 has now been updated to ease the interpretation of the pBT2436/pBT2101 GC content distribution.
- Given the megaplasmiids' low average GC content, atypical GC-composition islands mainly correspond to high GC regions (cyan peaks in Supplementary Fig. 1) that coincide with the location of AMR and transposase/integrase genes. An AT-rich island is also present in the pBT2436 RR1 region. Based on our new annotation of the IS identified in the pBT2436/pBT2101 resistance regions (see above), we show that these regions have been shaped by the collection of IS from different origins, with a plasmid from *Aeromonas salmonicida* being a recurrent match. Likewise, our comparative analysis of the region encoding the pBT2436 efflux pump (Supplementary Fig. 3), also rich in GC, suggest an origin from *Aeromonas hydrophila* but transfer among other bacterial species. We have added a sentence in the “The family of pBT2436-like megaplasmiids is widely distributed” results section to briefly acknowledge this:

“In pBT2436 and pBT2101, GC- and AT-rich regions coincided with the location of multiple transposase, integrase, and resistance genes predicted to be part of various IS with diverse taxonomic origins (Supplementary Fig. 1, Fig. 1).”

-A number of approaches have recently been applied to date the divergence of bacterial genomes, such as BEAST and BactDating, and it might be possible to use these approaches to generate a more refined estimate of the date of acquisition of multidrug resistance on these megaplasmiids

- We thank the reviewer for this suggestion and we have done our utmost to test its feasibility. This has involved a considerable amount of extra analysis. Since only 8 of the 13 complete megaplasmiid sequences compared in our study have an associated isolation date, critical for a dating analysis, we addressed this suggestion by first extending our comparative analysis to genomes that we identified from GenBank as carriers of pBT2436-like megaplasmiids. We compared all the proteins from these genomes to the core proteins detected by our pangenome analysis of the complete megaplasmiid sequences. We also compared our set of core proteins against the PAO1 gene products to ensure that matches identified against the GenBank genomes carrying megaplasmiids correspond to plasmid proteins only. No matches were found between the PAO1 and megaplasmiid core proteins. We restricted our comparison to genomes featuring matches that covered at least 80% of the pBT2436 sequence to maximize the identification of core proteins. Since many of the genomes from GenBank were not annotated, and in order to homogenize our

comparative analysis, we annotated all the analyzed genomes with the same pipeline prior comparison.

We then performed a series of tests for identifying the best phylogenetic markers among the core genes recovered from the new comparative analysis and inferred a maximum-Likelihood phylogeny under a Bayesian model using these markers. We finally used this phylogenetic tree and the reported dates of isolation from the GenBank genomes to conduct a dating analysis with the BactDating package.

The dating analysis found no statistical significance between the tree topology and the isolation dates associated to the analysed taxa ($R^2=0.01$, $p=2.12e-01$; see below), thus limiting the significance of the predicted nucleotide substitution rate (Rate= $3.58e-01$) and root date (MRCA=1879) of the rooted phylogenetic tree. Therefore, we decided not to include these results in the revised version of the manuscript. A number of factors such as the lack of a reported molecular clock for plasmids or the different taxonomic backgrounds of the analysed plasmids' host could account for the observed result.

Rate= $3.58e-01$,MRCA=1879.02, $R^2=0.01$, $p=2.12e-01$

Although the dating analysis did not render further insights into the megaplasמידs' evolution from a temporal point of view, the new comparative analysis and the corresponding phylogeny allowed us to considerably extend our knowledge on the megaplasמיד family diversity by connecting previously overlooked plasmids with their complete relatives, thus identifying new groups, confirming others, and highlighting abundant clusters. The new phylogenetic tree, now presented in a new Fig. 6, also reinforces our view on the dynamic nature of the megaplasמידs from both a taxonomic and geographic perspective, as it identifies more cases of plasmids from different species or countries clustering together. Additionally, the new comparative analysis allowed us to conclude that as few as four phylogenetic markers are sufficient to reproduce the tree topology inferred with hundreds of core genes, which has implications for the development of a megaplasמידs typing system.

These results are now presented in a paragraph part of the "Wider distribution of the pBT2436-like megaplasמיד family" results section, Fig. 6, Methods and Supplementary Methods:

"A phylogeny of the wider pBT2436-like megaplasמיד population, inferred from selected core gene sequences, revealed novel patterns of diversity previously unrecognized with the comparison of complete plasmids only (Fig 4 and 6). Although the overall topology of the two

phylogenetic trees is similar, we identified new clusters entirely formed by sequences recovered from our megaplasmid search in GenBank *Pseudomonas* genomes (Fig. 6). The most abundant groups were represented by the plasmids p1 and pOZ176 but several other sub-clusters and an apparent outlier were distinguished as well. Notably, only 4 phylogenetic markers were required to infer the tree suggesting that these genes could form the basis of a typing system.”

Reviewer #2 (Remarks to the Author):

General considerations

In this study, extensive bioinformatic searches have been used to characterize the structure and phylogeny of a number of megaplasmids from various geographical, environmental and clinical origins, some plasmids carrying multiple resistance genes to antibiotics, antiseptics and heavy metals. These data conclusively demonstrate that several megaplasmids hosted by *Pseudomonas* sp are closely or more distantly related, but belong to a same family, with a possible ancestor dating back to the 1970s or so. The manuscript is well written and scientifically sound.

Being not a specialist of bioinformatics, my comments will relate to other aspects of this paper.

1. If all the plasmids presented here are phylogenetically linked, intuitively one could assume that the most conserved part of their structure would include the genes that are essential for their replication, maintenance and partition. I suggest the authors provide and make a comment on this information.

- The reviewer’s assumption is correct, genes encoding functions associated to replication, transfer, and partition, among others, are part of the core genome. We indicate this in the “Core and accessory genome of the pBT2436-like megaplasmids” results section, Fig. 2 (outermost grey rings) and Supplementary Table 2 (full annotation):

“Based on the comparative analysis of the 15 members of the megaplasmid family, we identified a core genome consisting of 261 orthologous protein groups, including proteins with roles in plasmid replication and partitioning, plasmid transfer, heavy metal resistance, chemotaxis, and a set of SAM proteins (Fig. 2; Supplementary Table 2).”

- In this revised version of the manuscript, we additionally include an alignment of RepA protein sequences from the 15 complete megaplasmids to show the high level of conservation existing among them (Supplementary Figure 8).

2. Some of the plasmids presented carry genes of resistance to tellurite. Referring back to the pioneering works on *Pseudomonas* plasmids in the 1970s (e.g. by George Jacoby), large plasmids of incompatibility group IncP2 were considered as associated with tellurite resistance. Actually, nothing is said in the paper about the incompatibility group of this new family of replicons and whether all these contain the *ter* locus.

- As the reviewer points out, tellurite resistance genes were identified in the pBT2436-like plasmids, and these are indeed part of the core genome of the megaplasmids family as indicated in the Fig. 2 and Supplementary Table 2. Accordingly, one of the plasmids of the family (pOZ176) has been previously identified as IncP-2. We performed an alignment of RepA protein sequences from the complete megaplasmids that revealed a

high degree of sequence similarity among them (Supplementary Figure 8) and suggests that plasmids of this family belong to the same incompatibility group. This information is now indicated at the end of the “Core and accessory genome of the pBT2436-like megaplasmids” results section of the revised manuscript:

“Pseudomonas plasmids can be classified according to incompatibility group¹⁸. One member of the megaplasmid family, plasmid pOZ176 in *P. aeruginosa* PA96, was identified as IncP-2 using incompatibility testing methods¹⁹. Our genomics analysis revealed the presence of a conserved replication and stability system in the core backbone of the megaplasmid family (Supplementary Table 2). The RepA proteins share from 92 to 100% sequence identity (Supplementary Fig. 8) suggesting that all the members of the family belong to the same incompatibility group.

IncP-2 plasmids have been studied for many years, are considered ubiquitous in the environment, and are associated with tellurite resistance²⁰. We identified tellurite resistance genes (terABCDEFZ) as part of the megaplasmid family core genome (Supplementary Table 2).”

3. A number of the described plasmids contain widely distributed genes of resistance to different classes of antibiotics. Have attempts to transfer the plasmids to new hosts been attempted to provide further insights into their role in resistance gene diffusion, in correlation with their tra genes content ?

- Failed attempts to transfer the bla(IMP-45) marker from the *P. putida* strain SY153 carrying the megaplasmid pSY153-MDR to *E. coli* J53 and *P. aeruginosa* PAO1 by conjugation have been previously reported by Yuan et al (2017). However, in this revised version of our manuscript, we show that megaplasmids from *P. koreensis* and *P. aeruginosa* strains can be transferred by conjugation to *P. fluorescens* SBW25 at a high frequency. These transconjugants were selected by mobilisation of either kanamycin or mercury resistance and were confirmed by PCR. Thus, our new results indicate that the pBT2436-like megaplasmids conjugation machinery is functional and leads to inter-species diffusion of resistance genes. These data are presented in the new “Megaplasmids stability and fitness costs” results section, Fig 7, Supplementary Fig. 11, and Supplementary Table 6. Detailed methods corresponding to these experiments are presented in the “Methods” and “Supplementary Methods” sections. We have also added a co-author (James P.J. Hall), who contributed much of this additional data.

Minor points

4. Abstract and throughout the text : The word “pangenome” is used to describe the whole set of genes present in the considered megaplasmids. I think this is confusing since the notion of pangenome also includes non-plasmidic (chromosomal) genes.

- We have added a sentence in the “Methods” section of the revised manuscript to clarify that the pangenome analysis presented in this study refers only to the megaplasmids gene content:

“In this paper, we refer to pangenome analysis only in the context of the megaplasmid family, and not the chromosome.”

- When possible, we also replaced the term pangenome for plasmid core and/or accessory genome in order to avoid confusion.

5. A DNA fragment carrying a (nfxB)mexCD-oprJ-like locus has been found in several of these

megaplasmiids. It is not clear in the text (pages 4 and 7) how this fragment may have been collected from *A. hydrophila* (IS ?). A further comment on this would be useful.

- It is certainly unclear how the efflux pump was transferred to the pBT2436-like megaplasmiids. Our new annotation of Insertion Sequences (IS) (See above, Reviewer 1 - comment 1) in this region of pBT2436 did not reveal a recognizable IS in it, in contrast to the various elements identified in the pBT2436 and pBT2101 RR1 regions (Fig. 1, Supplementary Table 2). Still, putative integrase genes were found associated to the region containing the (nfxB)mexCD-oprJ-like locus, leaving the possibility open for the role of uncharacterized mobile elements in the mobilisation of the region (Fig. 1, Supplementary Fig. 3). Consistent with this idea, we found that regions displaying high sequence similarity in other plasmids mainly differ from this megaplasmiid region by the presence of genes associated with a range of different IS (Supplementary Fig. 3), confirming that mobile elements are commonly associated with this locus. We have now summarized this at the end of the “AMR regions in pBT2436 and pBT2101 are mosaic and dynamic” results section:

“Other close matches were found to regions of the non-related plasmids pBKPC18-1 from *Citrobacter freundii* (Accession CP022275) and pMKPA34-1 from *P. aeruginosa* (Accession MH547560) which mainly differed from the pBT2436 RR2 by the presence of genes associated with various IS, suggesting a role for these in its transmission (Supplementary Fig. 3).”

6. Page 4 (bottom), page 5 (top) and Table S3 please replace strain 3582 with strain 3583

- We thank the reviewer for spotting this mistake. We have now corrected this and checked for any other occurrences of the same error throughout the whole manuscript.

7. Page 11. EUCAST does not recommend to spread 10 microliters of a 10 McFarland bacterial suspension for the diffusion method, but rather to use a 0.5 McFarland suspension spread onto a Mueller-Hinton agar plate with a swab. Please, correct.

- We thank the reviewer for this correction . The method was written with incorrect details and we have now corrected this (Supplementary Methods):

“Antimicrobial susceptibility testing was carried out according to the EUCAST guidelines. Isolates to be tested were cultured onto Columbia plates (overnight 37°C). From these, single colonies were mixed with sterile distilled water to attain a standard optical density (0.5 McFarland units), and 10 µl spread onto Mueller-Hinton agar plates with a swab and incubated overnight at 37 °C with Meropenem (10 µg) ...”

Reviewer #3 (Remarks to the Author):

The authors report sequencing of two closely related *P. aeruginosa* megaplasmiids conferring MDR. Analysis of AMR regions revealed extensive variation, with various duplications and other rearrangements, highlighting the dynamic nature of these regions. Related plasmids were identified by querying publicly available sequence data, demonstrating a wide distribution of the megaplasmiid family across geographic locations, source types, and *Pseudomonas* species. The findings provide insight into plasmid-mediated AMR gene dissemination in *Pseudomonas*, and suggest that this megaplasmiid family may be an important vector for such spread. The manuscript is well written, thorough, and carefully thought through. I have only a few

suggestions for improvement:

1. Page 3: "In each of the three strains we identified circularised plasmid sequences." The text following implies that only a single plasmid was identified in each strain – is this correct? Please clarify wording.

- It is correct. In this work only one recognisable and circularised plasmid was identified per strain. 2436 and 2101 strain genomes were assembled into two circularised contigs, one corresponding to the chromosome and the other to the megaplasmid. The genome of 4068 genome could not have been assembled in one single contig and the only circularised contig from the assembly correspond to a plasmid. Although we cannot rule out the presence of additional plasmids in 4068 until the genome is completely assembled, we found that the other contigs show extensive sequence similarity to *P. aeruginosa* chromosomes but not to plasmids reported so far. As suggested by the reviewer we have now clarified the wording regarding this issue in the corresponding results section:

"Genomes of the strains 2436 and 2101 were assembled into two complete circularised contigs. 2436 and 2101 chromosomes are 6782092 and 6573638 bp long and feature 6214 and 6041 protein-coding genes, respectively. Both isolates also carried related megaplasms (named pBT2436 [423 kb] and pBT2101 [440 kb], respectively), harbouring multiple AMR genes."

"The genome of the strain 4068 was assembled into five contigs, one of which corresponded to a complete circularised plasmid of 51 kb with no identifiable AMR genes, therefore, it was not analysed further in this study."

2. Page 3: "...including a 35.9 kb region duplicated but inverted in pBT2101 when compared to pBT2436". I don't understand why pBT2436 is mentioned here. Isn't the duplication simply two copies of the same 35.9 kb sequence in opposite orientations in pBT2101?

- The reviewer is correct, here we refer to the orientation of the two 35.9 kb copies within the pBT2101 Resistance Region 1. We have corrected this sentence accordingly:

"By aligning each of the two larger Resistance Regions with themselves, we identified both large and small duplicated regions, including a 35.9 kb region duplicated but inverted in pBT2101, with a unique central region of 4.36 kb in between the two duplicated areas containing a mer operon (Fig. 1; Supplementary Fig. 2)."

3. Page 4: "A further example of how dynamic these regions are can be seen with the repeats 2b and 5b of pBT2436, located next to xerD (Fig. 1). Their pairs occupy different positions in pBT2436 RR1, whereas they occur merged as one repeat separated by 13 bp in the Resistance Region of pBT2101 (Fig. 1)" What exactly is meant by "different positions" in pBT2436? They look to be very close in the figure – how far apart are they exactly?

- We meant that the repeats 2b and 5b of pBT2436 are located next to each other in the Resistance Region 1 whereas their corresponding pairs are not; 2a and 5a are separated occupying distant positions within the same region. In contrast, this pair of repeats (2 and 5) occur as a single unit in pBT2101 designated as 3. Repeat 3 of pBT2101 has four copies (a, b, c and d), which are nearly identical to the pair 2b-5b of pBT2436 but include extra 13 bp in between. This case represents another example of the intricate nature of the rearrangements taking place in the megaplasmid Resistance

Regions, but we acknowledge it is difficult to describe and read. Hence we have decided to remove this part from the revised version of our manuscript.

4. Page 5: The finding that isolates 2101 and 3582, from different patients, have very similar plasmid sequences and almost identical MLST profiles is interesting. From the supplementary table it appears that the MLST profiles have 6/7 exact matches and one partial match – it is tempting to speculate that this partial match may be an assembly artefact rather than a genuine difference. Can the authors provide any additional insight here? Given the possible transmission link, I think it would be worthwhile investigating the relationship between these isolates at a more fine-grained level such as genome-wide SNVs. Also, in the discussion these isolates were described as “sharing the same MLST” – this description should be made consistent with the results section above.

- As the reviewer points out, one partial match is the difference that we found between the MLST profiles of the isolates 2101 and 3583. We inspected the alignments produced by the MLST profiling algorithm for the *guaA* locus where the partial match was detected and identified only one 1 bp mismatch between the two sequences. As the reviewer suggests this mismatch could be the result of an assembly artifact as the MLST profile was inferred from contig sequences. Hence we have now further assessed the similarity between 2101 and 3583, and the other Thai genomes, by performing a kmer-based clustering of all the sequences (new Supplementary Fig. 4). The resulting tree revealed that the diversity of the Thai clinical isolates is consistent with the population structure reported for *P. aeruginosa*, and confirmed the close relationship existing between the isolates carrying similar megaplasms, i.e. 2436 - 638 and 2101 - 3583. Although 2101 and 3583 are more closely related to each other than to any other genome in the collection, they still display divergence as compared to the strains 2436 and 638 which were isolated from different samples of the same patient. This observation is consistent with the variations identified in the 2101 and 3583 megaplasms, particularly the absence of a large duplication in the 3583 megaplasms. Although the high similarity between the 2101 and 3583 genomes suggests transmission between patients, the variation observed makes it difficult to support a recent event. We added some lines to the “Distribution of related megaplasms among clinical isolates” results section of the revised manuscript describing these findings:

“The multi locus sequence type (MLST) profiles extracted from the genomes and a kmer-based sequence comparison indicated that the isolates were highly diverse genetically (Supplementary Table 1 and Supplementary Fig. 4).”

“A phylogeny of the Thai isolates shows that the megaplasms can be found in strains of the two major *P. aeruginosa* groups (Supplementary Fig. 4). Isolates 2101 and 3583 are more closely related to each other than to any other isolate in the collection but they still display divergence compared to isolates 2436 and 638, obtained from the same patient. This is consistent with a transmission event linking the 2101 and 3583 isolates.”

- We have also fixed the corresponding sentence in the discussion as suggested:

“... and two from different patients but present in isolates sharing nearly identical MLST (isolates 2101 and 3583)”

5. Page 8: How genetically diverse were the matching samples from the GenBank genome search? It would be helpful to include MLST data or similar to get an idea of just how much

evidence there is for HGT.

- We have now MLST-profiled all the genomes from GenBank matching the pBT2436 plasmid and found a very diverse population featuring 33 different Sequence Types (STs) among the 56 *P. aeruginosa* genomes that were assigned to a particular ST (Supplementary Table 5, New Supplementary Fig. 10). The remaining genomes with no assigned ST under the *P. aeruginosa* scheme also showed different individual ST loci. The same observation was true for the non-*aeruginosa* genomes profiled under the *P. putida* or *P. fluorescens* MLST schemes. We did not observe a clear correlation between a ST and the samples country of isolation (Supplementary Fig. 10), nor did we identify any especially abundant STs suggesting the preference of the megaplasms for a particular genetic background. This finding, as pointed out by the reviewer, suggests a high level of Horizontal Gene Transfer mediated by the pBT2436-like megaplasms family and adds on our previous observations about the diversity of backgrounds that can host megaplasms. This finding is now summarized in the “Wider distribution of the pBT2436-like megaplasms family” results section of the manuscript:

“MLST profiles obtained from the megaplasms-carrier genomes portrayed a highly diverse population featuring 33 different Sequence Types (ST) among the 56 *P. aeruginosa* genomes assigned with a particular ST, thus indicating extensive horizontal gene transfer mediated by the plasmids (Supplementary Table 5 and Supplementary Fig. 10).”

6. Page 11: “increasing the number of open reading frames up to ~9%” => increasing the number of open reading frames by up to ~9%??

- We thank the reviewer for spotting this mistake. We have corrected this now.

7. The text in several of the figures is very small and difficult to read. I would suggest revisiting all the figures to ensure that minimum font sizes are maintained.

- We increased the font size in figures, where possible, while avoiding the labels overlapping. This change is best seen in Fig. 1. Supplementary figures were all improved regarding this issue as well.

8. Supplementary Fig. 1: What level of sequence identity is represented in the sequence alignment? Please clarify in the figure legend.

- The average sequence identity of the alignments is 98.43%. This is now indicated in the figure.

9. Data availability: For the PacBio genome sequences, I cannot access PRJNA540594 - please ensure this is made publicly available. For the Illumina sequences, it appears that some have already been published (Freschi et al 2019. *Genome Biol Evol.* 11(1):109-120). Please clarify which isolates were newly sequenced for this study. In addition, all raw read data (Illumina and PacBio) should be deposited in the SRA.

- The BioProject PRJNA540594 was made publicly available but the sequences set to be released upon publication. We have now requested the genomes to be released prior to publication, and both these and the PacBio sequencing reads should now be available from the referred BioProject accession number.

- We have now added a column in the Supplementary Table 1 to clearly indicate which sequences were reported in Freschi et al 2018.
- All Illumina and PacBio raw reads have now been submitted to SRA. The accession numbers are now indicated in the Supplementary Table 1 and the "Data availability" statement in the manuscript.

REVIEWERS' COMMENTS:

Reviewer #1 (Remarks to the Author):

The authors have adequately addressed the concerns that I raised over previous versions of this paper.

Reviewer #2 (Remarks to the Author):

All my previous remarks have been addressed satisfactorily by the authors and the manuscript in its revised version has been significantly improved.